# EFiGP: Eigen-Fourier Physics-Informed Gaussian Process for Inference of Dynamic Systems

## Abstract

Parameter estimation and trajectory reconstruction for data-driven dynamical systems governed by ordinary differential equations (ODEs) are essential tasks in fields such as biology, engineering, and physics. These inverse problems, estimating ODE parameters from observational data, are particularly challenging when the data are noisy, sparse, and the dynamics are nonlinear. We propose the Eigen-Fourier Physics-Informed Gaussian Process (EFiGP), an algorithm that integrates Fourier transformation and eigen-decomposition into a physics-informed Gaussian Process framework. This approach eliminates the need for numerical integration, significantly enhancing computational efficiency and accuracy. Built on a principled Bayesian framework, EFiGP incorporates the ODE system through probabilistic conditioning, enforcing governing equations in the Fourier domain while truncating high-frequency terms to achieve denoising and computational savings. The use of eigen-decomposition further simplifies Gaussian Process covariance operations, enabling efficient recovery of trajectories and parameters even in dense-grid settings. We validate the practical effectiveness of EFiGP on three benchmark examples, demonstrating its potential for reliable and interpretable modeling of complex dynamical systems while addressing key challenges in trajectory recovery and computational

## 1 Introduction

Systems of coupled Ordinary Differential Equations (ODEs) are essential tools for modeling the intricate mechanisms underlying various scientific and engineering processes, such as neuroscience (FitzHugh, 1961; Nagumo et al., 1962), ecology (Lotka, 1932), and systems biology (Hirata et al., 2002). We focus on dynamical systems governed by the following ODE formulation, as studied in Yang et al. (2021); Seifner et al. (2024); Gorbach et al. (2017):

$$\dot{\boldsymbol{x}}(t) = \frac{d\boldsymbol{x}(t)}{dt} = f(\boldsymbol{x}(t), \boldsymbol{\theta}, t), \quad t \in [0, T], \tag{1}$$

where the vector $\boldsymbol{x}(t)$ represents the system outputs that change over time $t$, and $\boldsymbol{\theta}$ is the vector of model parameters to be estimated from experimental or observational data, which is an ODE Inverse Problem (for a concise overview, see supplementary material §S2.1). When the function $f$ is nonlinear, determining $\boldsymbol{x}(t)$ given the initial conditions $\boldsymbol{x}(0)$ and $\boldsymbol{\theta}$ typically requires a numerical integration method, such as Runge–Kutta (Lapidus & Seinfeld, 1971).

Traditionally, ODEs have been utilized more for conceptual or theoretical insights rather than for fitting data, due to limitations in the availability of experimental data. However, advancements in experimental and data collection techniques have enhanced the ability to monitor dynamical systems in near real-time. Typically, such data are recorded at discrete time points and are subject to measurement errors. Therefore, we assume that the observations $\boldsymbol{y}(\boldsymbol{\tau}) = \boldsymbol{x}(\boldsymbol{\tau}) + \boldsymbol{\epsilon}(\boldsymbol{\tau})$ are made at $N$ specific time points $\boldsymbol{\tau} = (\tau_1, \tau_2, \ldots, \tau_N)$, with the error $\boldsymbol{\epsilon}(\boldsymbol{\tau})$ governed by a noise level $\sigma$. Our focus is on inferring the parameters $\boldsymbol{\theta}$ and recovering the ground-truth trajectory $\{\boldsymbol{x}(t)\}_{t=0}^{T}$ given data $\boldsymbol{y}(\boldsymbol{\tau})$, with particular emphasis on the nonlinear structure of $f$.

**Background** Bayesian inference and Gaussian Processes have long been utilized for calibrating parameters in dynamical systems (Kennedy & O'Hagan, 2001). More recently, MAnifold-

constrained Gaussian process Inference (MAGI) for ODEs (Yang et al., 2021) and Physics-Informed Gaussian Process (PIGP) for Partial Differential Equations (PDEs) (Li et al., 2024) have emerged as principled Bayesian approaches that inherently incorporate physical information to estimate parameters from observational data. These Bayesian counterparts to the Physics-Informed Neural Network (PINN) (Raissi et al., 2019) provide native uncertainty quantification within a theoretically rigorous Bayesian framework. MAGI and PIGP achieve this by leveraging a physics-informed Bayesian conditioning mechanism, which constrains the difference between derivative information obtained from the governing differential equations and that derived from a Gaussian Process (GP).

Focusing on ODEs, one of the key advantages of MAGI is that it bypasses numerical integration, leading to high computational efficiency with strong empirical performance. However, while the Bayesian conditioning of a physics-informed, ODE-driven manifold constraint provides a theoretically ideal inference method, its practical implementation requires discretization. This discretization introduces some degree of inaccuracy due to approximation errors. The discretized physics-informed constraint can be viewed as a collocation method, where the ODE information is conditioned only on specific collocation points. As a result, the computational burden of MAGI increases linearly with the number of discretization points. In this paper, we address this challenge by transforming the physics-informed, ODE-driven manifold constraint into Fourier space and applying spectral decomposition to the GP quadratic form. These techniques are employed to further reduce the computational cost associated with MAGI.

**Literature Review** Gaussian process (GP) surrogate modeling has been widely developed for inferring dynamical systems, offering a flexible, analytically tractable alternative within a Bayesian setting, as demonstrated by Hennig et al. (2015). The GP-based methods are including gradient matching approaches (Calderhead et al., 2008; Dondelinger et al., 2013; Ramsay, 2007; Wenk et al., 2019), physics-informed constraint methods (Huang et al., 2021; Zhao & Wong, 2024; Sun & Yang, 2023), Reproducing kernel Hilbert space (Chen et al., 2021), variational inference (Long et al., 2022), and probabilistic numerical methods (PNMs) (Kersting et al., 2020; Tronarp et al., 2022; Schmidt et al., 2021). Since PNMs are numerical solvers approaches, they fundamentally differ from both gradient matching and our proposed approach, as discussed in Kersting et al. (2020). Gradient matching methods typically define a joint GP over $\mathbf{x}$, $\dot{\mathbf{x}}$, and $\theta$ with hyperparameters $\phi$, yielding $p(\mathbf{x}, \dot{\mathbf{x}}, \theta) = p(\mathbf{x} \mid \theta)p(\dot{\mathbf{x}} \mid \mathbf{x}, \theta)$. Here, the GP prior governs $\dot{\mathbf{x}}$ while the ODE $\dot{\mathbf{x}} = f(\mathbf{x}, \theta)$ imposes a hard constraint. These two sources are not generally compatible, presenting a core limitation of gradient matching. The introduction of MAnifold-constrained Gaussian process Inference (MAGI) by Yang et al. (2021) and Wenk et al. (2020) addressed these challenges by resolving the theoretical incompatibilities of earlier GP-based approaches. MAGI not only improves the accuracy of parameter inference but also achieves computational efficiency, with its runtime scaling linearly with the number of system components (Wenk et al., 2020).

Physics-Informed Neural Network (PINN) (Raissi et al., 2019) offers another realm for solving differential equations using machine learning. By embedding the governing physical laws directly into the network's loss function, PINN can effectively handle high-dimensional and nonlinear PDEs without requiring large datasets, as the physics loss guides the optimization. However, despite their versatility, PINN can be computationally expensive and prone to failure, especially in multi-scale dynamical systems, due to challenges such as stiff gradients and sensitivity to hyperparameters (Wang et al., 2021). To address these limitations, Li et al. (2021) introduced the Fourier Neural Operator (FNO), which leverages Fourier transforms to solve differential equations by operating in the Fourier domain. In this space, differentiation simplifies multiplication, enabling FNO to efficiently capture long-range dependencies and complex interactions with quasi-linear time complexity. While FNO has demonstrated state-of-the-art approximation capabilities, they still require tens of thousands of training pairs generated by numerical solvers.

Given the computational limitations and performance instability of these approaches, there is a clear need for methods that are both robust and efficient. Notably, no prior work has explored incorporating Fourier transforms into Bayesian Gaussian Process frameworks that completely bypass numerical solvers, presenting an opportunity for innovation in this domain.

**Our Contribution** We propose a novel algorithm incorporating truncation with Fourier Transformation and Eigen-decomposition in the Physics-informed Gaussian Process (EFiGP). We demonstrate that the resulting parameter inference and trajectory recovery are statistically sound, compu-

tationally efficient, and effective in various practical scenarios. Our EFiGP not only addresses the limitations of previous Gaussian process approaches when discretization becomes very dense, but also improves accuracy and reduces computation time. A comprehensive discussion of computational efficiency is provided in Section S2.2. The code of EFiGP is provided on GitHub[1].

In particular, the physics information from the governing equation is enforced in the Fourier domain, which is especially useful for oscillatory ODEs that describe periodic or quasi-periodic systems. Examples include biological rhythms, such as the oscillations of Hes1 mRNA and Hes1 protein (Hirata et al., 2002), and relaxation oscillators, such as the FitzHugh-Nagumo equations (FitzHugh, 1961). Additionally, our approach allows for the truncation of high-frequency terms in the Fourier-transformed ODEs representation, achieving denoising while reducing computational costs.

The incorporation of Eigen-decomposition truncation in our algorithm enhances the computational efficiency and accuracy of parameter estimation and trajectory recovery. Eigen-decomposition allows us to diagonalize the covariance matrix of the Gaussian Process, which simplifies the computational complexity involved in the multiplication of large matrices. This is particularly advantageous in high-discretization settings, where points in recovered trajectories are highly correlated due to smoothness. By breaking down the Gaussian Process into orthogonal components and truncation, our approach efficiently captures the essential features of the dynamical system trajectories while discarding redundant information. This decomposition not only accelerates the computation but also enhances numerical stability, reducing the risk of errors due to ill-conditioned matrices.

## 2 PRELIMINARIES

### 2.1 MAGI: MANIFOLD-CONSTRAINED GAUSSIAN PROCESS INFERENCE

The MAGI framework, introduced by Yang et al. (2021), establishes a Bayesian approach to solve inverse problems using the Gaussian process. For a concise overview of the Gaussian process, see supplementary materials §S2.3. Within this Bayesian framework, the $D$-dimensional dynamical system $\boldsymbol{x}(t)$ is modeled as a realization of the stochastic process $\boldsymbol{X}(t) = (X_1(t), X_2(t), \ldots, X_D(t))$, with the model parameters $\boldsymbol{\theta}$ represented as realizations of the random variable $\boldsymbol{\Theta}$. The posterior distribution is then naturally derived. For clarity and conciseness, the main text omits the subscript $d$ corresponding to each dimension of the ODE system. For the complete $d$ notation, see §S2.4.

**Prior:** A general prior $\pi(\cdot)$ is imposed on $\boldsymbol{\theta}$, and an independent GP prior is assumed for each component $\boldsymbol{X}(t)$, such that

$$\boldsymbol{X}(t) \sim \mathcal{GP}(\mu, \mathcal{K}) \quad t \in [0, T], \tag{2}$$

where the mean function $\mu : \mathbb{R} \to \mathbb{R}$ and the positive-definite covariance function $\mathcal{K} : \mathbb{R} \times \mathbb{R} \to \mathbb{R}$ are parameterized by hyperparameters $\phi$. Therefore, for any finite set of time points $\boldsymbol{\tau}$, $\boldsymbol{X}(\boldsymbol{\tau})$ follows a multivariate Gaussian distribution with mean vector $\boldsymbol{\mu}(\boldsymbol{\tau})$ and covariance matrix $\mathcal{K}(\boldsymbol{\tau}, \boldsymbol{\tau})$.

**Likelihood:** Let the observations be denoted by $\boldsymbol{y}(\boldsymbol{\tau}) = (y(\tau_1), \ldots, y(\tau_N))$, where $\boldsymbol{\tau} = (\tau_1, \ldots, \tau_N)$ represents the set of $N$ observation time points for each component. For simplicity, the observation noise for each component is assumed to be i.i.d. zero-mean Gaussian with variance $\sigma^2$. Thus, the observation likelihood is given by:

$$\boldsymbol{Y}(\boldsymbol{\tau}) \mid \boldsymbol{X}(\boldsymbol{\tau}) = \boldsymbol{x}(\boldsymbol{\tau}) \sim \mathcal{N}(\boldsymbol{x}(\boldsymbol{\tau}), \sigma^2 \boldsymbol{I}_N) \tag{3}$$

**Physics Information:** A new random variable is introduced to quantify the difference between the time derivative $\dot{\boldsymbol{X}}(t)$ of the GP and the ODE structure for a given value of the parameter $\boldsymbol{\theta}$:

$$W = \sup_{t \in [0, T]} \left| \dot{\boldsymbol{X}}(t) - f(\boldsymbol{X}(t), \boldsymbol{\theta}, t) \right| \tag{4}$$

Under the event $\{W = 0\}$, the stochastic process $\boldsymbol{X}(t)$ fully satisfies ODE function Eq. 1. Therefore, conditioning on $W = 0$ will impose a physics-informed Bayesian constraint on the GP, $\boldsymbol{X}(t)$. However, since $W$ is a supremum over an uncountable set, it cannot be computed analytically. To address this, an approximation $W_I$ based on a finite discretization of the set $I = (t_1, t_2, \ldots, t_n)$ with $n$ discretization points is used, such that $\boldsymbol{\tau} \subset I \subset [0, T]$:

$$W_I = \left\{ \dot{\boldsymbol{X}}(t) - f(\boldsymbol{X}(t), \boldsymbol{\theta}, t) \right\}_{t \in I} \tag{5}$$

---

[1]https://anonymous.4open.science/r/EFiGP-F6AD

**Posterior:** The practically computable posterior distribution is given by

$$
\begin{aligned}
p_{\boldsymbol{\Theta}, \boldsymbol{X}(I) | W_I, \boldsymbol{Y}(\boldsymbol{\tau})} &(\boldsymbol{\theta}, \boldsymbol{x}(I) \mid W_I = \boldsymbol{0}, \boldsymbol{Y}(\boldsymbol{\tau}) = \boldsymbol{y}(\boldsymbol{\tau})) \\
&\propto \pi_{\boldsymbol{\Theta}}(\boldsymbol{\theta}) \times P(\boldsymbol{X}(I) = \boldsymbol{x}(I) \mid \boldsymbol{\Theta} = \boldsymbol{\theta}) \\
&\times P(\boldsymbol{Y}(\boldsymbol{\tau}) = \boldsymbol{y}(\boldsymbol{\tau}) \mid \boldsymbol{X}(I) = \boldsymbol{x}(I), \boldsymbol{\Theta} = \boldsymbol{\theta}) \\
&\times P(W_I = \boldsymbol{0} \mid \boldsymbol{Y}(\boldsymbol{\tau}) = \boldsymbol{y}(\boldsymbol{\tau}), \boldsymbol{X}(I) = \boldsymbol{x}(I), \boldsymbol{\Theta} = \boldsymbol{\theta}).
\end{aligned}
\tag{6}
$$

which represents the joint conditional distribution of $\boldsymbol{\theta}$ and $\boldsymbol{X}(I)$. The detailed formulation procedure is provided in §S2.5.

## 2.2 SPECTRAL DECOMPOSITION

To sample from a low-dimensional space for multivariate Gaussian distributions, the spectral decomposition (eigendecomposition) method can be applied.

**Lemma 2.1 (The Karhunen–Loeve theorem Stark & Woods (1986))** *Let $\Sigma$ be a covariance matrix with eigendecomposition $\Sigma = V \Lambda V^\top$, where $V$ is the matrix of eigenvectors, $\Lambda$ is the diagonal matrix of eigenvalues, and $J$ is the number of non-zero eigenvalues. Consider a random vector $\boldsymbol{Z} \sim \mathcal{N}(0, I_J)$, where $I_J$ is the $J \times J$ identity matrix. Then, the distribution of $\mu + V \Lambda^{1/2} \boldsymbol{Z}$ is $\mathcal{N}(\mu, \Sigma)$. This can be equivalently expressed as a sum:*

$$
\mu + \sum_{i=1}^{J} \sqrt{\lambda_i} z_i \boldsymbol{v}_i \sim \mathcal{N}(\mu, \Sigma),
\tag{7}
$$

*where $\lambda_i$ are the eigenvalues in $\Lambda$, $\boldsymbol{v}_i$ are the corresponding eigenvectors in $V$, and $z_i$ are the components of the random vector $\boldsymbol{Z}$.*

This formulation allows for truncation, where small eigenvalues are ignored to reduce computational complexity while maintaining a close approximation to the original distribution.

## 2.3 FOURIER TRANSFORMATION

Recall that the Discrete Fourier Transform (DFT) is a type of linear transformation that can be represented by a matrix, denoted as $A_{\text{DFT}}$. This matrix acts on vectors in $\mathbb{R}^n$, transforming them into vectors in $\mathbb{C}^n$. To work within a real-valued framework, we consider a linear mapping from $\mathbb{R}^n$ to $\mathbb{R}^{2n}$ that augments the DFT output by separating the real and imaginary components, represented by $\tilde{A}$. Specifically, this Fourier operator is constructed by combining the DFT matrix with a process that augments the output into its real and imaginary parts. By the properties of the Gaussian distribution, we have the following result:

**Lemma 2.2** *The DFT and augmentation of Gaussian random vector $\boldsymbol{X}(I) \sim \mathcal{N}(\boldsymbol{\mu}_I, K_{I,I})$ results in a multivariate Gaussian distribution. If we use $\mathcal{F}$ to denote Fourier transform and subsequent separation of real part and imaginary part, then $\mathcal{F}\{\boldsymbol{X}(I)\} = \tilde{A}\boldsymbol{X}(I) \sim \mathcal{N}(\tilde{A}\boldsymbol{\mu}_I, \tilde{A} \cdot K_{I,I} \cdot \tilde{A}^\top)$, where $\tilde{A}$ is the combined matrix form of the discrete Fourier transform and the augmentation process that separates the real and imaginary parts.*

Note that the resulting covariance matrix after applying this transformation is of dimension $\mathbb{R}^{2n-1 \times 2n-1}$, and the mean vector is in $\mathbb{R}^{2n-1}$, since the first imaginary component is zero. The detailed closed form of mapping matrix $\tilde{A}$ is included in the supplemental material §S2.6. This formulation also allows for truncation, where high-frequency terms are ignored to again reduce computational complexity while maintaining a close approximation to the original distribution.

## 3 EFiGP: EIGEN-FOURIER PHYSICS-INFORMED GAUSSIAN PROCESS

We tackle two limitations of the previous GP approach: When the discretization set becomes very dense, (1) the computational cost increases significantly, and (2) the algorithm may fail to converge to the ODEs solution trajectory due to highly correlated posterior samples. Thus, we combine the

ideas of eigen-decomposition and Fourier transformation to reduce computational cost in sampling and improve the accuracy of the random variable $W$ characterization of ODEs and GP discrepancy.

**Fourier:** We now measure the deviation of GP and the ODEs requirement in the Fourier space:

$$W_I^{\mathcal{F}} = \left\{ \mathcal{F}[\dot{\boldsymbol{X}}(I)] - \mathcal{F}[f(\boldsymbol{X}(I), \boldsymbol{\theta}, I)] \right\} \tag{8}$$

where the set $I = (t_1, t_2, \ldots, t_n)$ with $n$ discretization points. We can truncate the discrete Fourier series at the $l$-th term ($l < n$) to reduce the computational cost. Furthermore, we can easily obtain the computational form by Lemma 2.2 since $\dot{\boldsymbol{X}}$ is a joint Gaussian distribution (see Rasmussen & Williams (2006), chapter 9) as

$$P(W_I^{\mathcal{F}} = \boldsymbol{0} \mid \boldsymbol{Y}(\boldsymbol{\tau}) = \boldsymbol{y}(\boldsymbol{\tau}), \boldsymbol{X}(I) = \boldsymbol{x}(I), \boldsymbol{\Theta} = \boldsymbol{\theta})$$
$$= P(\mathcal{F}[\dot{\boldsymbol{X}}(I)] = \mathcal{F}[f(\boldsymbol{x}(I), \boldsymbol{\theta}, I)] \mid \boldsymbol{X}(I) = \boldsymbol{x}(I)) \tag{9}$$
$$\propto \exp\{-\frac{1}{2} \left\| \tilde{A}_{(l)}\{f_I^{\boldsymbol{\theta}, \boldsymbol{x}} - m\{\boldsymbol{x}(I) - \boldsymbol{\mu}(I)\}\} \right\|_{(C_{(l)}^{\mathcal{F}})^{-1}}^2 \}$$

where $f_I^{\boldsymbol{\theta}, \boldsymbol{x}}$ is short notation for $f(\boldsymbol{x}(I), \boldsymbol{\theta}, I)$, and $\| \cdot \|^2$ is short notation for quadratic form $\|\boldsymbol{r}\|_B^2 = \boldsymbol{r}^T B \boldsymbol{r}$. The matrix $\tilde{A}_{(l)}$ is the truncated Fourier transform matrix at $l$-th frequency term, and $C_{(l)}^{\mathcal{F}} = \tilde{A}_{(l)} \cdot C \cdot \tilde{A}_{(l)}^\top$ can be obtained by Lemma 2.2 on the conditional covariance matrix $C = \mathcal{K}''(I, I) -' \mathcal{K}(I, I)\mathcal{K}(I, I)^{-1}\mathcal{K}'(I, I)$, and $m =' \mathcal{K}(I, I)\mathcal{K}(I, I)^{-1}$. Finally, $'\mathcal{K} = \frac{\partial}{\partial s}\mathcal{K}(s, t)$, $\mathcal{K}' = \frac{\partial}{\partial t}\mathcal{K}(s, t)$, and $\mathcal{K}'' = \frac{\partial^2}{\partial s \partial t}\mathcal{K}(s, t)$. All the closed forms can be found in §S2.6.

**Eigen:** Since posterior sampling or maximum a posteriori (MAP) optimization on $\boldsymbol{X}(I)$ in Eq. 28 incurs a high cost when the set becomes denser, we propose an efficient way to handle $\boldsymbol{X}(I)$ by using spectral decomposition (Lemma 2.1). We consider the change of variable (orthogonally reparametrize) $\boldsymbol{X}(I)$ to $\boldsymbol{z} = (z_1, \ldots, z_n)$, using the matrix square root from the spectral decomposition of the prior variance and covariance matrix:

$$\boldsymbol{X}(I) = \boldsymbol{\mu}(I) + V_{(j)}\Lambda_{(j)}^{\frac{1}{2}}\boldsymbol{z} = \boldsymbol{\mu}(I) + \sum_{i=1}^j z_i \sqrt{\lambda_i}\boldsymbol{v}_i \tag{10}$$

where $\lambda_i, \boldsymbol{v}_i$ are eigenvalues and eigenvectors of $\mathcal{K}(I, I)$. The matrices $V_{(j)} = (\boldsymbol{v}_1, \ldots, \boldsymbol{v}_j)$, $\Lambda_{(j)} = \text{diag}(\lambda_1, \ldots, \lambda_j)$ are truncated eigen decomposition at $j$-th eigen value term. The $j$ is a hyperparameter that aims to save computational cost over all parts of the objective function Eq. 28.

**Posterior:** Now, our new practically computable posterior distribution for Eigen-Fourier Physics-Informed Gaussian Process (EFiGP) is:

$$P(\boldsymbol{\Theta} = \boldsymbol{\theta}, \boldsymbol{Z} = \boldsymbol{z} \mid W_I^{\mathcal{F}} = \boldsymbol{0}, \boldsymbol{Y}(\tau) = \boldsymbol{y}(\tau))$$
$$\propto P(\boldsymbol{\Theta} = \boldsymbol{\theta}, \boldsymbol{X}(I) = \boldsymbol{x}(I), W_I^{\mathcal{F}} = \boldsymbol{0}, \boldsymbol{Y}(\tau) = \boldsymbol{y}(\tau)) \times J(\boldsymbol{X}(I) \to \boldsymbol{Z})$$
$$\propto \pi_{\boldsymbol{\Theta}}(\boldsymbol{\theta}) \exp \left\{ -\frac{1}{2} \left[ \boldsymbol{z}^T \boldsymbol{z} + \left\| \tilde{A}_{(l)} \cdot f_I^{\boldsymbol{\theta}, \boldsymbol{x}} - \tilde{A}_{(l)} m\{V_{(j)}\Lambda_{(j)}^{\frac{1}{2}}\boldsymbol{z}\} \right\|_{(C_{(l)}^{\mathcal{F}})^{-1}}^2 \right. \right. \tag{11}$$
$$\left. \left. + \|\boldsymbol{\mu}(I) + V_{(j)}\Lambda_{(j)}^{\frac{1}{2}}\boldsymbol{z} - \boldsymbol{y}\|_{\sigma^{-2}}^2 \right] \right\}$$

where $\boldsymbol{x}(I) = \boldsymbol{\mu}(I) + V_{(j)}\Lambda_{(j)}^{\frac{1}{2}}\boldsymbol{z}$. The Jacobian $J(\boldsymbol{X}(I) \to \boldsymbol{Z})$ of the linear transformation is a constant that doesn't depend on $\boldsymbol{z}$ and therefore is dropped in the proportional sign. Eq. 11 is the computable-discretized posterior of EFiGP. In this paper, we consider the Maximum A Posteriori (MAP) as a fast point estimate from EFiGP, while the Posterior Mean and the Posterior Interval are the formal Bayesian inference results that further quantify the uncertainty.

## 4 SIMULATION RESULTS

In this section, we study the performance of EFiGP on three real-world systems: the FitzHugh-Nagumo (FN) (FitzHugh, 1961; Nagumo et al., 1962), the Lotka-Volterra (LV) (Lotka, 1932), and the Hes1 system (Hirata et al., 2002). We then compare our method with the vanilla Bayesian

GP method of MAGI, demonstrating that our proposed method improves the accuracy of inference results while significantly reducing run time.

**Data generation:** All ground truth data are simulated through numerical integration. Since FN, LV, and Hes1 are oscillators, we generate the true trajectories that cover approximately four to five cycles. To generate the noisy observations $y(\tau)$, we use 41 equally spaced data points from the first half period as training, covering about two cycles, with added i.i.d. Gaussian random noise. The second half period is reserved for out of sample prediction evaluation. Thus, only 41 observations are available for each component. We also assume that all components are observed at the observation time points, and we use the same noise level for all components. Note that the noise level is unknown when running our methodology.

**Benchmark models and Settings:** To the best of our knowledge, the MAGI framework has demonstrated better performance in previous comparisons with other inference methods based on GPs, such as adaptive gradient matching Wenk et al. (2019) and fast Gaussian process-based gradient matching Dondelinger et al. (2013). Therefore, **we evaluate our proposed EFiGP approach against the state-of-the-art MAGI** using varying discretization levels to show the effect on computational speed and accuracy. To further verify the robustness of our method, we also **compare our method with a differentiable classical ODE solver**. As this baseline is non-Bayesian, we include detailed information and the comparison in §S3.1. Lastly, since the initial guess may play a key role, we further examine the **sensitivity to the initial guess** for these methods and for a new benchmark method, `Fenrir`. For more information and the experimental results, see §S3.2. Regarding discretization, we use discretizations of $41, 81, \ldots, 1281$ equally spaced time points (e.g., 161 for $I = \{t_1, t_2, \ldots, t_{161}\}$). In our method, the hyperparameters include the GP kernel parameters, the noise level, and the truncation numbers of Fourier and Eigen. The GP kernel parameters and noise level are automatically tuned (see §S3.3 for more details). For the truncation numbers, we gradually increase the Fourier truncation $l$ and the eigen-decomposition truncation $j$ (e.g., 11, 21, etc.) until the results converge and stabilize. We also include an instruction for tuning these parameters (see §S3.4). For each system, we report the truncation numbers used at each discretization in Tab.S6 (FN system), Tab.S7 (Hes1 system), and Tab.S8 (LV system). Importantly, we observe that the **truncation numbers stabilize** at $j = 81$ and $l = 41$ as the discretization increases. These values represent the optimal trade-off between computational efficiency and accuracy for the systems under study. For the GP covariance function, we use the Matérn kernel with a degree of freedom of 2.01, ensuring that the kernel is twice differentiable.

**Evaluation Metric:** We evaluate model performance in recovering both the **true parameters and system trajectories**. For parameter estimation, we compute the **absolute error** between the inferred parameters and the pre-set true values. For each component's trajectory, we focus on the period of observation, together with one extended period of the same length that does not have any observation. Given that the observation time points differ from the discretization time points, we compute the **trajectory RMSE** over 2,561 predetermined equally spaced time points along the reconstructed trajectories. The reconstructed trajectory is generated via numerical integration, using the inferred initial condition $x_0$ (i.e., the first point of the inferred trajectory $x(I)$) and the inferred parameters. Notably, numerical integration is employed only for evaluation and forecasting in EFiGP and is not required for in-sample fitting.

## 4.1 Computational Complexity

In this section, we will discuss the computational complexity. After setting the GP kernel hyperparameters, we can pre-compute all covariance matrices, their eigendecompositions, and their corresponding Fourier-transformed bases. Without further approximation, each likelihood evaluation would require $O(n^2)$ work (e.g., dense matrix–vector multiplications), where $n$ is the number of time steps. However, following the computational efficiency techniques of Yang et al. (2021), we replace each covariance matrix with a banded approximation, reducing dense $O(n^2)$ computations to $O(bn)$, where $b$ is the fixed bandwidth. Additionally, we apply dimensional reductions: an eigen-decomposition truncation to dimension $j$, and a Fourier-domain truncation to dimension $l$. The resulting computational cost involves mapping $z$ from dimension $j$ to $2l - 1$, costing $O(jl)$, and performing matrix–vector multiplications, costing $O(bl)$. Since $j, b, l$ are all fixed dimensions, significantly smaller than $n$ and independent of $n$, our computational cost is substantially lower than that of the original MAGI approach.

Our analysis is consistent with the running times shown in Tab.1. From this table, we can observe that the average runtime of EFiGP over 100 repetitions remains constant across all discretization sizes for each system. For the FN system, EFiGP is approximately six times faster than MAGI at a discretization size of 1281. For the Hes1 system, after a discretization size of 161, EFiGP becomes twice as fast as MAGI. For a discretization size of 641, EFiGP is about four times faster, and at 1281, it is approximately six times faster. For the LV system, after a discretization size of 321, EFiGP becomes twice as fast as MAGI. At a discretization size of 1281, EFiGP is approximately 5 times faster than MAGI. In contrast, the runtime of MAGI increases almost linearly as the discretization becomes denser.

Table 1: Computational cost comparison between MAGI and EFiGP on different systems with different discretization levels, based on 100 repetitions. All computational times (in seconds) are measured on a 2017 MacBook Pro

| Discretization | FN System | | Hes1 System | | LV System | |
|---|---|---|---|---|---|---|
| | EFiGP | MAGI | EFiGP | MAGI | EFiGP | MAGI |
| 41 | 8.02±0.81 | 11.0±3.20 | 9.39±2.08 | 20.5±3.38 | 9.91±1.40 | 15.1±2.18 |
| 81 | 7.93±0.67 | 11.2±2.22 | 10.4±2.99 | 20.8±2.51 | 8.11±0.59 | 13.3±2.16 |
| 161 | 6.44±0.68 | 9.66±0.34 | 9.12±0.65 | 14.4±0.79 | 7.07±1.32 | 10.1±0.18 |
| 321 | 7.04±0.55 | 11.7±0.54 | 9.80±0.75 | 17.8±2.75 | 6.84±0.39 | 11.8±0.31 |
| 641 | 7.62±0.31 | 17.7±1.36 | 10.3±0.67 | 28.5±2.85 | 7.79±0.34 | 18.6±1.68 |
| 1281 | 7.41±0.44 | 39.1±1.23 | 10.5±0.36 | 62.4±4.60 | 7.59±0.44 | 39.6±1.02 |

## 4.2 PERFORMANCE EVALUATION

In this section, we present experimental results and compare performance with MAGI on three ODE systems. Although we primarily evaluate our method on oscillatory ODEs in this study, we also apply it to **non-oscillatory systems**, where it achieves slightly better accuracy and delivers a substantial reduction in runtime. Since this lies somewhat beyond our main focus, we include an experiment on the **chaotic Lorenz model in the SI** (see §S3.5).

Table 2: Mean and standard deviations of RMSE for MAGI and EFiGP for each component on the LV, FN, and Hes1 systems

| System | Component | Method | 41 | 81 | 161 | 321 | 641 | 1281 |
|---|---|---|---|---|---|---|---|---|
| FN | $x_1$ | EFiGP | 0.70±0.38 | 0.89±0.36 | 0.21±0.05 | 0.28±0.14 | 0.31±0.13 | 0.28±0.12 |
| | | MAGI | 0.42±0.28 | 0.48±0.21 | 0.29±0.11 | 0.30±0.13 | 0.39±0.15 | 0.43±0.15 |
| | $x_2$ | EFiGP | 0.26±0.16 | 0.34±0.16 | 0.09±0.04 | 0.10±0.04 | 0.11±0.04 | 0.09±0.04 |
| | | MAGI | 0.17±0.12 | 0.26±0.09 | 0.22±0.06 | 0.20±0.04 | 0.20±0.05 | 0.21±0.04 |
| Hes1 | $\log(x_1)$ | EFiGP | 0.32±0.15 | 0.24±0.09 | 0.19±0.06 | 0.17±0.04 | 0.12±0.03 | 0.09±0.02 |
| | | MAGI | 0.30±0.16 | 0.22±0.09 | 0.21±0.12 | na | na | na |
| | $\log(x_2)$ | EFiGP | 0.23±0.12 | 0.17±0.08 | 0.11±0.04 | 0.07±0.02 | 0.09±0.02 | 0.11±0.02 |
| | | MAGI | 0.22±0.12 | 0.15±0.07 | 0.12±0.07 | na | na | na |
| | $\log(x_3)$ | EFiGP | 0.64±0.28 | 0.47±0.17 | 0.37±0.13 | 0.34±0.08 | 0.21±0.05 | 0.18±0.05 |
| | | MAGI | 0.59±0.29 | 0.43±0.18 | 0.38±0.19 | na | na | na |
| LV | $\log(x_1)$ | EFiGP | 0.16±0.04 | 0.13±0.07 | 0.10±0.06 | 0.06±0.03 | 0.04±0.03 | 0.06±0.02 |
| | | MAGI | 0.17±0.12 | 0.12±0.09 | 0.09±0.05 | 0.06±0.03 | 0.06±0.03 | 0.11±0.05 |
| | $\log(x_2)$ | EFiGP | 0.23±0.06 | 0.18±0.10 | 0.15±0.08 | 0.08±0.04 | 0.05±0.03 | 0.06±0.02 |
| | | MAGI | 0.25±0.18 | 0.18±0.14 | 0.12±0.09 | 0.08±0.05 | 0.06±0.04 | 0.11±0.07 |

Table 3: Mean and standard deviations of Absolute Error for MAGI and EFiGP for each parameter on the FN system

| | | 41 | 81 | 161 | 321 | 641 | 1281 |
|---|---|---|---|---|---|---|---|
| **EFiGP** | a | .020±.013 | .025±.022 | .027±.026 | .030±.026 | .031±.025 | .031±.024 |
| | b | .176±.116 | .121±.099 | .190±.111 | .257±.114 | .264±.112 | .233±.103 |
| | c | .100±.075 | .174±.094 | .068±.047 | .075±.051 | .067±.046 | .050±.034 |
| **MAGI** | a | .026±.019 | .017±.014 | .028±.017 | .033±.017 | .032±.017 | .031±.018 |
| | b | .120±.082 | .229±.141 | .388±.144 | .475±.123 | .500±.099 | .500±.085 |
| | c | .126±.083 | .241±.105 | .288±.104 | .277±.097 | .251±.087 | .231±.080 |

**FN system:**

The FitzHugh-Nagumo (FN) system was introduced by FitzHugh (1961); Nagumo et al. (1962) to model the activation of excitable systems such as neurons. It is a two-component system governed by the following ODEs:

$$\begin{cases} \dot{x}_1 = c\left(x_1 - \frac{x_1^3}{3} + x_2\right), \\ \dot{x}_2 = -\frac{x_1 - a + bx_2}{c}, \end{cases} \quad (12)$$

where $a = 0.2$, $b = 0.2$, $c = 3$, and $x(0) = (-1, 1)$ are the true parameter values and initial conditions. We simulated 100 datasets with a noise level of 0.2 across both components. These parameter values and initial conditions are used to generate the ground-truth trajectory. Fig.S3 visualizes one example dataset and evaluation period by using EFiGP and MAGI.

In terms of performance, EFiGP consistently yields more accurate results across the two components as the discretization increased, along with improved parameter estimation accuracy compared to MAGI. EFiGP has the most outperformance when the discretization is dense enough at 161. As seen in Tab. 2 and Tab. 3, EFiGP yields more stable results for each component and improves parameter estimation accuracy as the discretization increases beyond 161 (four times denser). Specifically, the EFiGP stabilizes at 161, and further increasing the discretization size beyond 321 does not further improve or degrade the results. On the contrary, the MAGI results on $x_1$ deteriorate as the discretization increases beyond 321.

**Hes1 system:** The Hes1 system was introduced by Hirata et al. (2002) to model the oscillatory dynamics of the Hes1 protein level ($x_1$) and Hes1 mRNA level ($x_2$) under the influence of a Hes1-interacting factor ($x_3$). It is a three-component system governed by the following ODEs:

$$\begin{cases} \dot{x}_1 = -ax_1x_3 + bx_2 - cx_1, \\ \dot{x}_2 = -dx_2 + \frac{e}{1+x_1^2}, \\ \dot{x}_3 = -ax_1x_3 + \frac{f}{1+x_1^2} - gx_3, \end{cases} \quad (13)$$

where the true parameter values are $a = 0.022$, $b = 0.3$, $c = 0.031$, $d = 0.028$, $e = 0.5$, $f = 20$, and $g = 0.3$. The initial condition is $x(0) = (1.438575, 2.037488, 17.90385)$. These parameter values and initial conditions are used to generate the ground-truth trajectory. Since the Hes1 system variables are strictly positive, we apply a log transformation to each component. We simulated 100 datasets with a log-normal noise of 0.1, using 41 observations. Fig.S4 visualizes one example dataset in log scale together with reconstructed trajectories on the fitting period and prediction period by using EFiGP and MAGI.

Table 4: Mean and standard deviations of Absolute Error for MAGI and EFiGP for each parameter on the Hes1 system

|  |  | 41 | 81 | 161 | 321 | 641 | 1281 |
|---|---|---|---|---|---|---|---|
| **EFiGP** | a | 0.002±0.001 | 0.001±0.001 | 0.001±0.001 | 0.001±0.001 | 0.001±0.001 | 0.001±0.001 |
|  | b | 0.024±0.018 | 0.023±0.017 | 0.022±0.019 | 0.039±0.031 | 0.059±0.039 | 0.071±0.043 |
|  | c | 0.004±0.003 | 0.004±0.002 | 0.003±0.002 | 0.004±0.003 | 0.007±0.004 | 0.008±0.005 |
|  | d | 0.001±0.001 | 0.001±0.001 | 0.001±0.001 | 0.003±0.002 | 0.005±0.002 | 0.005±0.002 |
|  | e | 0.026±0.030 | 0.024±0.016 | 0.026±0.018 | 0.035±0.028 | 0.082±0.041 | 0.112±0.045 |
|  | f | 10.156±0.092 | 10.254±0.079 | 10.279±0.085 | 10.296±0.089 | 10.304±0.089 | 10.315±0.086 |
|  | g | 0.194±0.019 | 0.191±0.019 | 0.167±0.022 | 0.153±0.023 | 0.152±0.023 | 0.151±0.023 |
| **MAGI** | a | 0.002±0.001 | 0.001±0.001 | 0.002±0.003 | na | na | na |
|  | b | 0.028±0.020 | 0.023±0.017 | 0.028±0.045 | na | na | na |
|  | c | 0.004±0.003 | 0.004±0.002 | 0.004±0.005 | na | na | na |
|  | d | 0.001±0.001 | 0.001±0.001 | 0.001±0.002 | na | na | na |
|  | e | 0.026±0.017 | 0.025±0.016 | 0.037±0.055 | na | na | na |
|  | f | 10.142±0.099 | 10.247±0.081 | 10.279±0.089 | na | na | na |
|  | g | 0.195±0.018 | 0.189±0.019 | 0.167±0.024 | na | na | na |

Tab.2 summarizes the accuracy of the reconstructed trajectories for the three system components, while Tab.4 reports the estimation accuracy of the parameters. Both tables present results across different discretization levels. In general, EFiGP demonstrates improved performance in trajectory reconstruction as the discretization level increases up to 641. Meanwhile, parameter estimation accuracy stabilizes at a lower discretization level of 161. This may be attributed to better recovery of weakly identifiable parameter combinations that deviate from the true values but yield similar trajectories. In contrast, MAGI fails to converge when the discretization exceeds 321, underscoring the enhanced robustness of EFiGP.

**LV system:** The Lotka-Volterra (LV) system was introduced by Lotka (1932) to model the dynamics of predator-prey interactions. It is a two-component system governed by the following ODEs:

$$\begin{cases} \dot{x}_1 = ax_1 - bx_1x_2, \\ \dot{x}_2 = cx_1x_2 - dx_2, \end{cases} \quad (14)$$

where $a = 1.5$, $b = 1$, $c = 1$, and $d = 3$ are the true parameter values, and $x(0) = (5, 0.2)$ is the initial condition. These parameters are used to generate the ground-truth trajectory. Since the Hes1 system variables are strictly positive, we apply a log transformation to each component. We simulated 100 datasets with a log-normal noise of 0.1, using 41 observations. Fig.S5 visualizes one example dataset in exponential scale together with reconstructed trajectories on the fitting period and prediction period by using EFiGP and MAGI.

Table 5: Mean and standard deviations of Absolute Error for MAGI and EFiGP for each parameter on the LV system with a tuned learning rate

|       |   | 41 | 81 | 161 | 321 | 641 | 1281 |
|-------|---|----|----|-----|-----|-----|------|
| **EFiGP** | a | .026±.019 | .026±.019 | .036±.025 | .054±.028 | .067±.028 | .078±.024 |
|       | b | .027±.019 | .027±.020 | .033±.025 | .049±.029 | .056±.028 | .040±.026 |
|       | c | .028±.019 | .029±.019 | .036±.025 | .057±.028 | .067±.027 | .073±.019 |
|       | d | .051±.032 | .049±.034 | .066±.044 | .102±.053 | .121±.055 | .159±.034 |
| **MAGI** | a | .028±.023 | .026±.021 | .035±.024 | .056±.028 | .079±.029 | .119±.035 |
|       | b | .028±.022 | .027±.021 | .031±.022 | .049±.027 | .058±.029 | .042±.029 |
|       | c | .022±.019 | .022±.018 | .032±.021 | .058±.025 | .069±.025 | .072±.022 |
|       | d | .045±.036 | .044±.035 | .059±.042 | .105±.052 | .139±.053 | .170±.050 |

Tab. 2 and Tab. 5 present the performance of EFiGP and MAGI across varying discretization levels. EFiGP consistently demonstrates more stable trajectory reconstruction across all components, with accuracy improving as the discretization increases beyond 321 (eight times denser). Notably, EFiGP outperforms MAGI in trajectory accuracy for components $x_1$ and $x_2$ at higher discretization levels. However, at finer discretization levels beyond 161, EFiGP's parameter estimation accuracy deteriorates, despite continued improvements in trajectory reconstruction. Further analysis reveals that due to issues with differentiability, parameter combinations that deviate more from the ground truth can produce trajectories nearly indistinguishable from the actual ones, as shown in SI Fig.S6.

## 5 Discussion and Conclusion

In this paper, we introduce a methodology for inferring dynamical systems using eigen-decomposed, Fourier-transformed, and physics-informed Gaussian Process. The Fourier transform provides several key advantages over working in the original space: (1) Incorporating physics information in the Fourier domain averages discrepancies in derivative information between the ODEs and the GP across the entire domain, rather than limiting it to discretization points. (2) Adding more frequency terms progressively introduces orthogonal information, while increasing the number of discretization points often leads to diminishing returns due to growing correlations. (3) For oscillatory ODEs, enforcing physics information in the Fourier domain ensures long-term reliability, whereas discretization points may fail to generalize beyond their coverage. Additionally, our method achieves better computational efficiency and accuracy as the discretization size increases. It outperforms existing GP-based approaches in inference accuracy on benchmark examples, with significantly faster computation times.

However, parameter estimation accuracy still degrades with increased discretization, largely due to the weak identifiability inherent in ODE systems. Fortunately, this issue can be mitigated by providing more observations, as shown in §S3.1. However, designing methods to further address this limitation is an interesting direction for future research. Moreover, while this study focuses on point estimation, future work should explore uncertainty quantification within the Bayesian framework. This would enhance the robustness of our method and enable recovery of the full range of plausible weakly identifiable parameters. We also acknowledge that our method offers only modest accuracy improvements while substantially reducing computational cost. Addressing this limitation also represents an interesting direction for future research.

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

## S1  LLMs Statement

In this paper, we used an LLM (i.e., ChatGPT) only for proofreading (grammar, typos, and sentence-level clarity). It was not used for research ideation or substantive writing.

## S2  Technical Appendices

### S2.1  ODE Inverse Problem

Typically, an ODE model is expressed as Eq. 1. In many practical applications, we are faced with the challenge of determining the underlying parameters $\boldsymbol{\theta}$ from observed data. This leads to the ODE inverse problem, which can be formally stated as follows. Given a set of observed data points $\{(\tau_i, y(\tau_i))\}_{i=1}^{N}$, determine the parameter $\boldsymbol{\theta}$ such that the solution $\boldsymbol{x}(\boldsymbol{\tau}; \boldsymbol{\theta})$ of the ODE in Eq. 1 best fits the observed data in the sense of minimizing the discrepancy between the ODE solution and the observations. The inverse problem is often approached through optimization techniques, where an objective function, usually the sum of squared differences between the observed and the ODE solution, is minimized:

$$\min_{\boldsymbol{\theta}} \sum_{i=1}^{N} \|x(\tau_i; \boldsymbol{\theta}) - y(\tau_i)\|^2.$$

Each evaluation of loss function typically requires solving $x(\tau_i; \boldsymbol{\theta})$ using numerical integration.

### S2.2  Computational Efficiency

The computational cost of our approach comprises two components: precomputed matrix operations (overhead) and the optimization procedure. Following the original MAGI framework, once the kernel hyperparameters are fixed, which are automatically tuned by our method (see , certain matrices can be formed once and then reused throughout optimization. This reduces posterior inference to simple matrix multiplications, avoiding repeated expensive operations such as eigendecomposition and Fourier transforms, and thus significantly improves efficiency. Table S1 reports the averaged overhead cost across discretizations of 161, 321, 641, and 1281 points for both MAGI and EFiGP on the Hes1 system. As shown, EFiGP incurs slightly higher overhead than MAGI. We only present the highest-resolution result here, since the matrix computation cost depends only on the number of components and discretization size.

Table S1: Averaged computational cost (in seconds) across discretizations

| Method | 161 | 321 | 641 | 1281 |
|--------|------|------|------|------|
| EFiGP  | 0.103 | 0.363 | 1.794 | 6.736 |
| MAGI   | 0.134 | 0.464 | 1.331 | 5.068 |

### S2.3  GP Smoothing

Gaussian Process (GP) smoothing is a powerful non-parametric technique used to model and predict complex, noisy data. Formally, a Gaussian Process is specified by its mean function $m(\boldsymbol{\tau})$, which is often assumed to be zero for simplicity, and its covariance function $k(\boldsymbol{\tau}, \boldsymbol{\tau}')$, which defines the relationship between different points in the input space. For any finite set of points $\boldsymbol{\tau} = (\tau_1, \tau_2, \ldots, \tau_N)$, we have:

$$\boldsymbol{X}(\boldsymbol{\tau}) \sim \mathcal{GP}(m(\boldsymbol{\tau}), k(\boldsymbol{\tau}, \boldsymbol{\tau}')),$$

Given a set of observed data points $\{(\tau_i, y_i)\}_{i=1}^{N}$, where $\boldsymbol{Y}(\boldsymbol{\tau}) = \boldsymbol{X}(\boldsymbol{\tau}) + \boldsymbol{\epsilon}$ and $\epsilon_i \sim \mathcal{N}(0, \sigma^2)$ represents noise, GP smoothing aims to infer the function $\boldsymbol{X}(\boldsymbol{\tau})$. The joint distribution of the

observed outputs $\boldsymbol{Y}(\boldsymbol{\tau})$ and the predicted values $\boldsymbol{X}(\tilde{\boldsymbol{\tau}})$ at test points $\tilde{\boldsymbol{\tau}} = (\tilde{\tau}_1, \tilde{\tau}_2, \ldots, \tilde{\tau}_m)$ is given by:

$$
\begin{bmatrix} \boldsymbol{Y}(\boldsymbol{\tau}) \\ \boldsymbol{X}(\tilde{\boldsymbol{\tau}}) \end{bmatrix} \sim \mathcal{N} \left( 0, \begin{bmatrix} K(\boldsymbol{\tau}, \boldsymbol{\tau}) + \sigma^2 I & K(\boldsymbol{\tau}, \tilde{\boldsymbol{\tau}}) \\ K(\tilde{\boldsymbol{\tau}}, \boldsymbol{\tau}) & K(\tilde{\boldsymbol{\tau}}, \tilde{\boldsymbol{\tau}}) \end{bmatrix} \right),
$$

where $K(\boldsymbol{\tau}, \boldsymbol{\tau})$ is the covariance matrix evaluated at the training points, $K(\boldsymbol{\tau}, \tilde{\boldsymbol{\tau}})$ is the covariance between the training and test points, and $K(\tilde{\boldsymbol{\tau}}, \tilde{\boldsymbol{\tau}})$ is the covariance matrix at the test points. The posterior distribution over the function values at the test points $\boldsymbol{X}(\tilde{\boldsymbol{\tau}})$, given the observed data, is:

$$
\boldsymbol{X}(\tilde{\boldsymbol{\tau}}) \mid \boldsymbol{\tau}, \boldsymbol{Y}(\boldsymbol{\tau}), \tilde{\boldsymbol{\tau}} \sim \mathcal{N}(\tilde{\mu}, \tilde{\mathrm{cov}}),
$$

where the mean and covariance of the posterior distribution are given by:

$$
\tilde{\mu} = K(\tilde{\boldsymbol{\tau}}, \boldsymbol{\tau}) \left[ K(\boldsymbol{\tau}, \boldsymbol{\tau}) + \sigma^2 I \right]^{-1} \boldsymbol{Y}(\boldsymbol{\tau}),
$$

$$
\tilde{\mathrm{cov}} = K(\tilde{\boldsymbol{\tau}}, \tilde{\boldsymbol{\tau}}) - K(\tilde{\boldsymbol{\tau}}, \boldsymbol{\tau}) \left[ K(\boldsymbol{\tau}, \boldsymbol{\tau}) + \sigma^2 I \right]^{-1} K(\boldsymbol{\tau}, \tilde{\boldsymbol{\tau}}).
$$

GP smoothing provides not only predictions but also measures of uncertainty, making it a robust method for modeling and interpreting noisy data. Its applications span various fields, including geostatistics, machine learning, and time-series analysis.

### S2.4 FULL $d$ NOTATION

**Prior:** We impose a general prior $\pi(\cdot)$ on $\theta$ and an independent GP prior on each component $\boldsymbol{X}_d(t)$:

$$
\boldsymbol{X}_d(t) \sim \mathcal{GP}(\boldsymbol{\mu}_d, \boldsymbol{\mathcal{K}}_d) \quad t \in [0, T] \tag{15}
$$

where the mean function $\boldsymbol{\mu}_d : \mathbb{R} \to \mathbb{R}$ and the positive-definite covariance function $\boldsymbol{\mathcal{K}}_d : \mathbb{R} \times \mathbb{R} \to \mathbb{R}$ are parameterized by hyperparameters $\phi_d$.

**Likelihood:** For any finite set of time points $\boldsymbol{\tau}_d$, $\boldsymbol{X}_d(\boldsymbol{\tau}_d)$ has a multivariate Gaussian distribution:

$$
\boldsymbol{Y}_d(\boldsymbol{\tau}_d) \mid \boldsymbol{X}_d(\boldsymbol{\tau}_d) = \boldsymbol{x}_d(\boldsymbol{\tau}_d) \sim \mathcal{N}(\boldsymbol{x}_d(\boldsymbol{\tau}_d), \sigma_d^2 \boldsymbol{I}_{N_d}) \tag{16}
$$

We define the random variable $W$ quantifying the difference between the time derivative $\dot{\boldsymbol{X}}_d(t)$ of the GP and the ODE structure:

$$
W = \sup_{d=1,\ldots,D; t \in [0,T]} \left| \dot{\boldsymbol{X}}_d(t) - f(\boldsymbol{X}_d(t), \theta, t) \right| \tag{17}
$$

Since $W$ cannot be computed analytically, we approximate it with $W_I$ using a finite discretization:

$$
W_I = \max_{d=1,\ldots,D; t \in I} \left| \dot{\boldsymbol{X}}_d(t) - f(\boldsymbol{X}_d(t), \theta, t) \right| \tag{18}
$$

**Posterior:** The computable posterior distribution is:

$$
p_{\boldsymbol{\Theta}, \boldsymbol{X}_d(I) | W_{I,d}, \boldsymbol{Y}_d(\boldsymbol{\tau}_d)}(\theta, \boldsymbol{x}_d(I) \mid W_{I,d} = 0, \boldsymbol{Y}_d(\boldsymbol{\tau}_d) = \boldsymbol{y}_d(\boldsymbol{\tau}_d)) \tag{19}
$$

By Bayes' rule, we have:

$$
\begin{aligned}
&p_{\boldsymbol{\Theta}, \boldsymbol{X}_d(I) | W_{I,d}, \boldsymbol{Y}_d(\boldsymbol{\tau}_d)}(\theta, \boldsymbol{x}_d(I) \mid W_{I,d} = 0, \boldsymbol{Y}_d(\boldsymbol{\tau}_d) = \boldsymbol{y}_d(\boldsymbol{\tau}_d)) \propto \\
&P(\Theta = \theta, \boldsymbol{X}_d(I) = \boldsymbol{x}_d(I), W_{I,d} = 0, \boldsymbol{Y}_d(\boldsymbol{\tau}_d) = \boldsymbol{y}_d(\boldsymbol{\tau}_d))
\end{aligned} \tag{20}
$$

The closed form of the right-hand side is:

$$
\begin{aligned}
&P(\Theta = \theta, \boldsymbol{X}_d(I) = \boldsymbol{x}_d(I), W_{I,d} = 0, \boldsymbol{Y}_d(\boldsymbol{\tau}_d) = \boldsymbol{y}_d(\boldsymbol{\tau}_d)) \\
&= \pi_{\Theta}(\theta) \times P(\boldsymbol{X}_d(I) = \boldsymbol{x}_d(I) \mid \Theta = \theta) \\
&\quad \times P(\boldsymbol{Y}_d(\boldsymbol{\tau}_d) = \boldsymbol{y}_d(\boldsymbol{\tau}_d) \mid \boldsymbol{X}_d(I) = \boldsymbol{x}_d(I), \Theta = \theta) \\
&\quad \times P(W_{I,d} = 0 \mid \boldsymbol{Y}_d(\boldsymbol{\tau}_d) = \boldsymbol{y}_d(\boldsymbol{\tau}_d), \boldsymbol{X}_d(I) = \boldsymbol{x}_d(I), \Theta = \theta)
\end{aligned} \tag{21}
$$

The ODE information part of EFiGP is

$$W_I^{\mathcal{F}} = \max_{d=1,\ldots,D; t\in I} \left| \mathcal{F}[\dot{\boldsymbol{X}}_d(t)] - \mathcal{F}[f(\boldsymbol{X}_d(t), \theta, t)] \right| \tag{22}$$

Where the set $I = (t_1, t_2, \ldots, t_n)$ with $n$ discretization points. Also, we can easily obtain the computational form by Lemma 2.2 since $\dot{\boldsymbol{X}}_d$ is a joint Gaussian distribution.

Secondly, since posterior sampling or maximum a posteriori (MAP) optimization on $\boldsymbol{X}_d(I)$ in the objective function (28) incurs a high cost when the set becomes denser, we propose an efficient way to handle $\boldsymbol{X}_d(I)$ by using spectral decomposition (Lemma 2.1). We consider the change of variable (orthogonally reparametrize) $\boldsymbol{X}_d(I)$ to $\boldsymbol{z}_d = (z_{d1}, \ldots, z_{dn})$, using the matrix square root from the spectral decomposition of the prior variance and covariance matrix:

$$\boldsymbol{X}_d(I) = \boldsymbol{\mu}_d(I) + \boldsymbol{V}_{d(j)}\boldsymbol{\Lambda}_{d(j)}^{\frac{1}{2}}\boldsymbol{z}_d = \boldsymbol{\mu}_d(I) + \sum_{i=1}^{j} z_{di}\sqrt{\lambda_{di}}\boldsymbol{v}_{di} \tag{23}$$

where $\lambda_{di}, \boldsymbol{v}_{di}$ are eigenvalues and eigenvectors of $\mathcal{K}_d(I, I)$. By this form, we can also truncate the summation and keep the first $j$ terms to save computational cost over all parts of the objective function (28).

**Objective Function of EFiGP with full $d$:**

$$P(\Theta = \theta, \boldsymbol{Z}_d = \boldsymbol{z}_d \mid W_{I,d}^{\mathcal{F}} = 0, \boldsymbol{Y}_d(\tau_d) = \boldsymbol{y}_d(\tau_d))$$

$$\propto P(\Theta = \theta, \boldsymbol{X}_d(I) = \boldsymbol{x}_d(I), W_{I,d}^{\mathcal{F}} = 0, \boldsymbol{Y}_d(\tau_d) = \boldsymbol{y}_d(\tau_d)) \times J(\boldsymbol{X}_d(I) \rightarrow \boldsymbol{Z}_d)$$

$$\propto \pi_\Theta(\theta)\exp\{-\frac{1}{2}\left[|I|\log(2\pi) + \boldsymbol{z}_d^T\boldsymbol{z}_d\right. \tag{24}$$

$$+ |I|\log(2\pi) + \log|\boldsymbol{K}_{d,k}^{\mathcal{F}}| + \left\|\mathcal{F}[f_{d,I}^{\theta,\boldsymbol{x}}]_k - \mathcal{F}[m_d\{\boldsymbol{V}_{d(j)}\boldsymbol{\Lambda}_{d(j)}^{\frac{1}{2}}\boldsymbol{z}_d\}]_k\right\|_{(\boldsymbol{K}_{d,k}^{\mathcal{F}})^{-1}}^2$$

$$\left. + N_d\log(2\pi\sigma_d^2) + \|\boldsymbol{V}_{d(j)}\boldsymbol{\Lambda}_{d(j)}^{\frac{1}{2}}\boldsymbol{z}_d(\tau_d) - \boldsymbol{y}_d(\tau_d)\|_{\sigma_d^{-2}}^2\right]\}$$

where $\boldsymbol{V}_{d(j)} = (\boldsymbol{v}_{d1}, \ldots, \boldsymbol{v}_{dj})$, $\boldsymbol{\Lambda}_{d(j)} = \text{diag}(\lambda_{d1}, \ldots, \lambda_{dj})$, and $\boldsymbol{K}_{d,k}^{\mathcal{F}}$ can be obtained by property with the truncated number $k \in \mathbb{N}$. The (integral) Jacobian of the linear transformation is a constant that doesn't depend on $\boldsymbol{z}_d$ and therefore is dropped in the proportional sign. Also, the short notations are:

$$\|\boldsymbol{v}_d\|_A^2 = \boldsymbol{v}_d^T A\boldsymbol{v}_d$$

$$m_d =' \mathcal{K}_d(I, I)\mathcal{K}_d(I, I)^{-1}$$

$$K_d = \mathcal{K}_d''(I, I) -' \mathcal{K}_d(I, I)\mathcal{K}_d(I, I)^{-1}\mathcal{K}_d'(I, I)$$

where $'\mathcal{K}_d = \frac{\partial}{\partial s}\mathcal{K}_d(s, t)$, $\mathcal{K}_d' = \frac{\partial}{\partial t}\mathcal{K}_d(s, t)$, and $\mathcal{K}_d'' = \frac{\partial^2}{\partial s\partial t}\mathcal{K}_d(s, t)$.

After optimizing $\boldsymbol{z}_d$, we can transfer it back to the original space with a dense discretization by:

$$\boldsymbol{X}_d(I) = \boldsymbol{\mu}_d(I) + \boldsymbol{V}_{d(j)}\boldsymbol{\Lambda}_{d(j)}^{\frac{1}{2}}\boldsymbol{z}_d \tag{25}$$

## S2.5 POSTERIOR OF MAGI

The practically computable posterior distribution is given by

$$p_{\Theta, \boldsymbol{X}(I)|W_I, \boldsymbol{Y}(\tau)}(\boldsymbol{\theta}, \boldsymbol{x}(I) \mid W_I = 0, \boldsymbol{Y}(\tau) = \boldsymbol{y}(\tau)) \tag{26}$$

which represents the joint conditional distribution of $\boldsymbol{\theta}$ and $\boldsymbol{X}(I)$. This formulation allows for the simultaneous inference of both the parameters $\boldsymbol{\theta}$ and the unobserved trajectory $\boldsymbol{X}(I)$ from the noisy observations $\boldsymbol{y}(\tau)$. Using Bayes' rule, this posterior can be expressed as:

$$p_{\Theta, \boldsymbol{X}(I)|W_I, \boldsymbol{Y}(\tau)}(\boldsymbol{\theta}, \boldsymbol{x}(I) \mid W_I = 0, \boldsymbol{Y}(\tau) = \boldsymbol{y}(\tau)) \propto$$

$$P(\Theta = \boldsymbol{\theta}, \boldsymbol{X}(I) = \boldsymbol{x}(I), W_I = 0, \boldsymbol{Y}(\tau) = \boldsymbol{y}(\tau)). \tag{27}$$

The right-hand side can be expressed in closed form as follows:

$$
\begin{aligned}
P(\boldsymbol{\Theta} &= \boldsymbol{\theta}, \boldsymbol{X}(I) = \boldsymbol{x}(I), W_I = 0, \boldsymbol{Y}(\boldsymbol{\tau}) = \boldsymbol{y}(\boldsymbol{\tau})) \\
&= \pi_{\boldsymbol{\Theta}}(\boldsymbol{\theta}) \times P(\boldsymbol{X}(I) = \boldsymbol{x}(I) \mid \boldsymbol{\Theta} = \boldsymbol{\theta}) \\
&\times P(\boldsymbol{Y}(\boldsymbol{\tau}) = \boldsymbol{y}(\boldsymbol{\tau}) \mid \boldsymbol{X}(I) = \boldsymbol{x}(I), \boldsymbol{\Theta} = \boldsymbol{\theta}) \\
&\times P(W_I = 0 \mid \boldsymbol{Y}(\boldsymbol{\tau}) = \boldsymbol{y}(\boldsymbol{\tau}), \boldsymbol{X}(I) = \boldsymbol{x}(I), \boldsymbol{\Theta} = \boldsymbol{\theta}).
\end{aligned}
\tag{28}
$$

### S2.6 CLOSED FORM OF TRANSFORMATION MATRIX

Given $\boldsymbol{X} \sim \mathcal{N}(0, \boldsymbol{\Sigma})$, where $\boldsymbol{X} \in \mathbb{R}^n$ and $\boldsymbol{\Sigma} \in \mathbb{R}^{n \times n}$, and a Discrete Fourier Transform (DFT) matrix, $\boldsymbol{A}$, we can write $\boldsymbol{Y} \in \mathbb{C}^n$ as:

$$
\boldsymbol{Y} = \boldsymbol{A}\boldsymbol{X} = \boldsymbol{U} + i\boldsymbol{V},
$$

where $\boldsymbol{U}$ and $\boldsymbol{V}$ are the real and imaginary parts of $\boldsymbol{Y}$, respectively. We can then define the augmented vector $\tilde{\boldsymbol{Y}}$ as:

$$
\tilde{\boldsymbol{Y}} = \begin{pmatrix} \boldsymbol{U} \\ \boldsymbol{V} \end{pmatrix},
$$

and there is a matrix $\tilde{\boldsymbol{A}} = \begin{pmatrix} \Re(\boldsymbol{A}) \\ \Im(\boldsymbol{A}) \end{pmatrix}$ such that:

$$
\tilde{\boldsymbol{Y}} = \tilde{\boldsymbol{A}}\boldsymbol{X},
$$

where $\Re(\boldsymbol{A})$ and $\Im(\boldsymbol{A})$ are the operators that extract the real and imaginary components, respectively. The distribution of $\tilde{\boldsymbol{Y}}$ can be expressed as:

$$
\tilde{\boldsymbol{Y}} \sim \mathcal{N}\left(0, \boldsymbol{\Sigma}_{\tilde{Y}}\right),
$$

where

$$
\boldsymbol{\Sigma}_{\tilde{Y}} = \begin{bmatrix} \frac{1}{2}\Re(\boldsymbol{A}\boldsymbol{\Sigma}\boldsymbol{A}^*) + \frac{1}{2}\Re(\boldsymbol{A}\boldsymbol{\Sigma}\boldsymbol{A}^T) & -\frac{1}{2}\Im(\boldsymbol{A}\boldsymbol{\Sigma}\boldsymbol{A}^*) + \frac{1}{2}\Im(\boldsymbol{A}\boldsymbol{\Sigma}\boldsymbol{A}^T) \\ \frac{1}{2}\Im(\boldsymbol{A}\boldsymbol{\Sigma}\boldsymbol{A}^*) + \frac{1}{2}\Im(\boldsymbol{A}\boldsymbol{\Sigma}\boldsymbol{A}^T) & \frac{1}{2}\Re(\boldsymbol{A}\boldsymbol{\Sigma}\boldsymbol{A}^*) - \frac{1}{2}\Re(\boldsymbol{A}\boldsymbol{\Sigma}\boldsymbol{A}^T) \end{bmatrix}.
$$

## S3 EXPERIMENTAL APPENDICES

### S3.1 COMPARISON WITH ODE SOLVER

In this section, we compare our `EFiGP` with a differentiable classical ODE solver, the Runge–Kutta method, as a strong baseline (McGreivy & Hakim, 2024). Given data $\mathcal{D} = \{y(\boldsymbol{\tau})\}$ observed at $N$ time points on the grid $\boldsymbol{\tau} \subset I$ such that $\boldsymbol{\tau} = \{\tau_1, \tau_2, \dots, \tau_N\}$, the Runge–Kutta method minimizes the loss over $D$ component system $\boldsymbol{x}(t) = (x_1(t), x_2(t), \dots, x_D(t))$ as

$$
L = \sum_{d \in D} \sum_{\tau_i \in \boldsymbol{\tau}} \|y_d(\tau_i) - x_d(\tau_i)\|_2^2,
\tag{29}
$$

where $\boldsymbol{x}(t)$ is computed using a classical Runge-Kutta initial value solver. In this experiment, we use the `Tsit5` solver (Tsitouras, 2011), initialized with a step size of 0.01 under adaptive step-size control. The implementation is provided by `diffrax` (Kidger, 2021). Since $\boldsymbol{x}(t)$ is always obtained from the ODE solver, the discretization size does not play a key role here. Hence, we compare our `EFiGP` with the numerical solver (denoted as `RK`) under different setups by increasing the number of observations $\boldsymbol{\tau}$. To evaluate our `EFiGP` comprehensively under a new experimental setup, we also consider `MAGI` as a benchmark. Due to convergence issues with the JAX optimizer and the inability of MAGI to converge at a discretization size of 1281 for the Hes1 system, we instead evaluate our method on the FN and LV systems. In this experiment, we consider observation sizes $\tau \in \{41, 81, 161, 321\}$. For both `EFiGP` and `MAGI`, the discretization size is fixed at 1281. In our paper, we consider a noise level of 0.2 for the FN system and a log-normal noise level of 0.1 for the LV system. We then simulated 20 datasets with the setup described before (i.e., initial values and true parameters) .

From the Tab.S2, we observe that the performance of inferred trajectories improves as the number of observations increases for all methods. However, our `EFiGP` consistently outperforms `MAGI` in

both parameter estimation and trajectory recovery. With 41 observations, the `RK` method achieves an RMSE on $x_1$ that is roughly half that of `EFiGP`. Yet, with 321 observations, the inferred trajectories become comparable to those obtained from the `RK` method. More importantly, for the inferred parameter $c$, our `EFiGP` provides more accurate estimates than `RK`. For the LV system, the results are shown in Tab.S3, and we observe the same pattern, with the performance of inferred trajectories improving as the number of observations increases. Even with only 41 observations, the recovered trajectory for the first component is comparable to that obtained with the `RK` method. Considering the trade-off between performance and computational cost as shown in Tab.S4, we conclude that `EFiGP` can provide the results at the same precision level while saving computational cost (i.e., faster by an order of magnitude)

More importantly, as we showed in the previous section, we found that the performance degradation is due to weak identifiability inherent in ODE systems. Fortunately, our method fully resolves this issue on the Hes1 system (e.g., `MAGI` fails to converge as the discretization ($I$) increases). From this section, we can see that increasing the number of observations helps mitigate the issue.

Table S2: Mean $\pm$ standard deviations of Absolute Error (for parameters) and RMSE (for trajectories) for MAGI, EFiGP, and Numerical Solver (RK) on the FN system.

|  |  | 41 | 81 | 161 | 321 |
|---|---|---|---|---|---|
| **EFiGP** | a | 0.031±0.024 | 0.027±0.016 | 0.015±0.008 | 0.017±0.006 |
|  | b | 0.233±0.103 | 0.126±0.019 | 0.099±0.025 | 0.086±0.016 |
|  | c | 0.050±0.034 | 0.027±0.014 | 0.024±0.008 | 0.019±0.008 |
|  | $x_1$ | 0.276±0.123 | 0.122±0.042 | 0.104±0.041 | 0.077±0.014 |
|  | $x_2$ | 0.091±0.038 | 0.041±0.008 | 0.037±0.007 | 0.034±0.010 |
| **MAGI** | a | 0.031±0.018 | 0.036±0.013 | 0.020±0.013 | 0.018±0.011 |
|  | b | 0.500±0.085 | 0.458±0.034 | 0.408±0.049 | 0.343±0.050 |
|  | c | 0.231±0.080 | 0.209±0.045 | 0.156±0.044 | 0.129±0.050 |
|  | $x_1$ | 0.427±0.153 | 0.285±0.069 | 0.237±0.042 | 0.184±0.044 |
|  | $x_2$ | 0.208±0.037 | 0.170±0.014 | 0.154±0.027 | 0.126±0.020 |
| **RK** | a | 0.041±0.023 | 0.019±0.017 | 0.022±0.015 | 0.016±0.011 |
|  | b | 0.126±0.010 | 0.109±0.016 | 0.051±0.040 | 0.047±0.031 |
|  | c | 0.038±0.022 | 0.021±0.017 | 0.023±0.020 | 0.026±0.014 |
|  | $x_1$ | 0.127±0.076 | 0.084±0.034 | 0.062±0.035 | 0.057±0.025 |
|  | $x_2$ | 0.064±0.029 | 0.033±0.008 | 0.022±0.004 | 0.020±0.004 |

Table S3: Mean and standard deviations of Absolute Error (for parameters) and RMSE (for trajectories) for MAGI, EFiGP, and Numerical Solver (RK) on the LV system

|  |  | 41 | 81 | 161 | 321 |
|---|---|---|---|---|---|
| **EFiGP** | a | 0.078±0.024 | 0.094±0.012 | 0.055±0.009 | 0.034±0.005 |
|  | b | 0.040±0.026 | 0.047±0.036 | 0.041±0.004 | 0.029±0.340 |
|  | c | 0.073±0.019 | 0.070±0.009 | 0.046±0.008 | 0.032±0.002 |
|  | d | 0.159±0.034 | 0.185±0.038 | 0.097±0.025 | 0.075±0.008 |
|  | $\log(x_1)$ | 0.064±0.019 | 0.056±0.011 | 0.043±0.009 | 0.038±0.002 |
|  | $\log(x_2)$ | 0.058±0.024 | 0.049±0.011 | 0.048±0.016 | 0.041±0.002 |
| **MAGI** | a | 0.119±0.035 | 0.172±0.012 | 0.118±0.013 | 0.066±0.015 |
|  | b | 0.042±0.029 | 0.095±0.033 | 0.031±0.002 | 0.017±0.007 |
|  | c | 0.072±0.022 | 0.130±0.010 | 0.083±0.008 | 0.046±0.005 |
|  | d | 0.170±0.050 | 0.356±0.034 | 0.174±0.024 | 0.084±0.021 |
|  | $\log(x_1)$ | 0.113±0.042 | 0.103±0.012 | 0.098±0.008 | 0.067±0.020 |
|  | $\log(x_2)$ | 0.107±0.063 | 0.078±0.011 | 0.074±0.034 | 0.074±0.031 |
| **RK** | a | 0.460±0.046 | 0.457±0.037 | 0.470±0.014 | 0.437±0.010 |
|  | b | 0.021±0.011 | 0.050±0.029 | 0.041±0.009 | 0.030±0.012 |
|  | c | 0.018±0.022 | 0.020±0.014 | 0.015±0.007 | 0.013±0.008 |
|  | d | 0.090±0.036 | 0.081±0.064 | 0.048±0.023 | 0.061±0.016 |
|  | $\log(x_1)$ | 0.056±0.023 | 0.043±0.017 | 0.034±0.009 | 0.032±0.007 |
|  | $\log(x_2)$ | 0.034±0.008 | 0.026±0.006 | 0.023±0.008 | 0.020±0.004 |

Table S4: Runtime (i.e., optimization time) comparison of RK and EFiGP on FN and LV systems across different observation sizes, based on 20 repetitions. All computational times (in seconds) are measured on a MacBook M3 chip

| Observations | FN System | | LV System | |
|---|---|---|---|---|
| | RK | EFiGP | RK | EFiGP |
| 41 | 11.37±0.41 | 1.84±0.01 | 11.24±0.61 | 1.99±0.01 |
| 81 | 11.47±0.26 | 1.84±0.02 | 10.64±0.59 | 2.09±0.06 |
| 161 | 11.35±0.41 | 1.96±0.08 | 11.03±0.11 | 2.10±0.05 |
| 321 | 11.28±0.57 | 2.00±0.05 | 11.49±0.11 | 2.13±0.08 |

## S3.2 SENSITIVE ANALYSIS

In this section, we consider a more realistic scenario from the user's point of view. Given a dataset for a known system, the user may not know any information about the parameters, but only the initial condition (i.e., $x(t_0)$). However, in the ODE inverse problem, the user should provide an initial guess for those parameters. We recognize that some methods require a good initial guess to obtain meaningful results; otherwise, the algorithm may diverge. Therefore, we designed the following experiment to explore the sensitivity to the initial guess and to demonstrate which method is easier to use.

In this experiment, we examine the sensitivity of the initial guess using the FN system and the LV system with three methods: EFiGP, Fenrir, and RK. Fenrir is a probabilistic numerical method and the code is provided by probdiffeq (Kr"amer, 2024). For the FN system, we simulated a single dataset with a noise level of 0.2 across both components, using 41 observations. We follow the same procedure for the LV system, but use a log-normal noise level of 0.1 and 161 observations. Since the primary goal is exploring the sensitivity to the initial guess, we fix the dataset and vary only the initial guess. After fixing the dataset, we define the sampling range as $\{\tilde{\theta}\} = \{\theta \pm \alpha\delta\}$, where $\delta \sim \mathcal{N}(0,1)$, $\alpha$ is a scaling factor, and $\{\theta\}$ are the ground-truth ODE parameters. We randomly simulated 50 initial guesses from this range. Next, we applied all methods to each initial guess and computed the average RMSE across components (i.e., the mean of the RMSE of $x_1$ and the RMSE of $x_2$). An algorithm is considered to have failed if the average RMSE $\geq 1$; otherwise, it is considered successful. The success rate (i.e., success/total) is then computed for all methods. For the FN system, we set $\alpha = 2$, yielding parameter ranges of $a = 0.2 \pm 2$, $b = 0.2 \pm 2$, and $c = 3 \pm 2$. For the LV system, we set $\alpha = 0.5$. The truncation numbers are fixed at $j = 81$ and $l = 41$ for both systems.

Table S5: Success rate for the FN system over 50 random initial guess.

| | EFiGP | RK | Fenrir |
|---|---|---|---|
| FN system | 44/50 | 35/50 | 33/50 |
| LV system | 49/50 | 41/50 | 40/50 |

From Tab.S5, we can see that EFiGP achieves a higher success rate compared with the numerical solver based methods for both systems. In the small range case, the success rate of EFiGP is about 98%. Among the numerical solvers, the RK method performs slightly better than Fenrir. Furthermore, the success rate increases significantly when the initial guess is closer to the ground-truth parameter. Overall, these results indicate that EFiGP is easier to use without requiring a precise initial guess.

## S3.3 TUNING THE HYPERPARAMETERS

For the hyperparameters of the GP kernel and noise level $\sigma$, we automatically tune by our method. Mathematically, we placed a GP prior on each component $X_d(t)$ as $X_d(t) \sim \mathcal{GP}(\mu_d, \mathcal{K}_d)$, $t \in [0, T]$. We use the Matérn kernel

$$\mathcal{K}(I) = \phi_1 \frac{2^{1-\nu}}{\Gamma(\nu)} \left(\sqrt{2\nu}\,\frac{I}{\phi_2}\right)^\nu B_\nu\left(\sqrt{2\nu}\,\frac{I}{\phi_2}\right),$$

which has two hyperparameters that are held fixed during sampling: $\phi_1$ controls the overall variance level of the GP, while $\phi_2$ controls the bandwidth (i.e. how much neighboring points of the GP affect each other). The values of $(\phi_1, \phi_2, \sigma)$ are obtained jointly by maximizing the GP marginal likelihood without conditioning on any ODE information:

$$(\widetilde{\phi}, \widetilde{\sigma}) = \arg\max_{\phi_1, \phi_2, \sigma} p\big(\phi_1, \phi_2, \sigma^2 \mid y_{I_0}\big) = \arg\max_{\phi_1, \phi_2, \sigma} \pi_{\phi_1}(\phi_1)\,\pi_{\phi_2}(\phi_2)\,\pi_\sigma(\sigma^2)\,p\big(y_{I_0} \mid \phi_1, \phi_2, \sigma^2\big),$$

where $y_{I_0} \mid \phi, \sigma \sim \mathcal{N}\big(0, \mathcal{K}_\phi + \sigma^2\big)$. The index set $I_0$ is the smallest evenly spaced set such that all observation time points in this component lie in $I_0$ (i.e., $\tau \subseteq I_0$). The priors $\pi_{\Phi_1}(\phi_1)$ and $\pi_\sigma(\sigma^2)$ for the variance parameters $\phi_1$ and $\sigma$ are set to be flat. The prior $\pi_{\Phi_2}(\phi_2)$ for the bandwidth parameter $\phi_2$ is set to be a Gaussian distribution.

### S3.4    TABLE FOR TRUNCATION NUMBER

As we have mentioned in paper, we gradually increase the Fourier series $l$ and spectral decomposition terms $j$ (e.g., 11, 21, 41, etc.) until results converge and stabilize. We have attached heatmaps showing the averaged RMSE across components to determine the optimal truncation numbers for the case of 1281 discretization. Note that the decision is based on two decimal place precision. In the heatmap, $z_t$ denotes the eigen-decomposition truncation number, and $k$ denotes the Fourier truncation number. From all the heatmaps, we observe that performance stabilizes at $j = 81$ and $l = 41$. Since this is for illustrative purposes, the heatmap shows the averaged RMSEs for all combinations of truncation numbers. To efficiently tune the truncation numbers, we recommend the following two-step approach:

1. Tune $j$ first with a low $l$. Performance improves most rapidly as users increase $j$ in this regime; in our experiments on both systems, RMSE stabilized after about three to four increments of $j$.

2. Tune $l$. With $j$ fixed, adjusting $l$ is straightforward and yields further fine-grained gains.

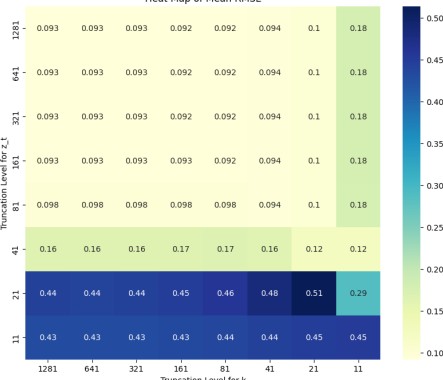

Figure S1: the Hes1 system

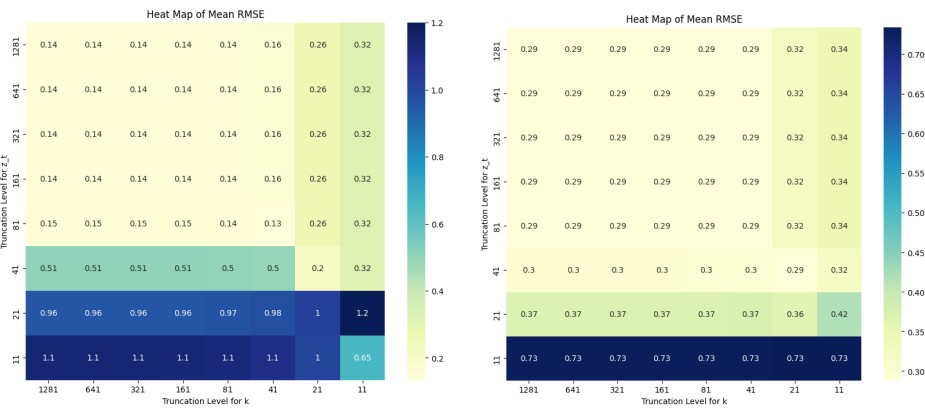

Figure S2: LHS is the FN system. RHS is the LV system

We investigated the truncation numbers for E (eigen-decomposition) and F (fourier transformation) and summarized the optimal values across all discretization sizes for all systems.

Table S6: FitzHugh-Nagumo (FN) System

| Discretization | 41 | 81 | 161 | 321 | 641 | 1281 |
|---|---|---|---|---|---|---|
| E | 41 | 41 | 81 | 81 | 81 | 81 |
| F | 11 | 11 | 21 | 41 | 41 | 41 |

Table S7: Hes1 System

| Discretization | 41 | 81 | 161 | 321 | 641 | 1281 |
|---|---|---|---|---|---|---|
| E | 21 | 81 | 81 | 81 | 81 | 81 |
| F | 11 | 21 | 21 | 41 | 41 | 41 |

Table S8: Lotka-Volterra (LV) System

| Discretization | 41 | 81 | 161 | 321 | 641 | 1281 |
|---|---|---|---|---|---|---|
| E | 41 | 41 | 81 | 81 | 81 | 81 |
| F | 21 | 21 | 41 | 41 | 41 | 41 |

## S3.5 CHAOTIC LORENZ SYSTEM

Table S9: Performance comparison across discretizations

| Method | Metric | 161 | 321 | 641 | 1281 |
|---|---|---|---|---|---|
| MAGI | Time | $4.2603 \pm 0.2260$ | $5.7201 \pm 0.3142$ | $7.8567 \pm 0.4716$ | $16.4346 \pm 0.4942$ |
| | $\sigma$ | $0.9566 \pm 0.0037$ | $2.3319 \pm 0.0015$ | $2.7240 \pm 0.0007$ | $2.8842 \pm 0.0004$ |
| | $\rho$ | $1.2118 \pm 0.0030$ | $1.1967 \pm 0.0031$ | $0.8119 \pm 0.0034$ | $0.3378 \pm 0.0026$ |
| | $\beta$ | $0.2004 \pm 0.0010$ | $0.2019 \pm 0.0009$ | $0.2976 \pm 0.0008$ | $0.3984 \pm 0.0007$ |
| EFiGP (e=161, f=81) | Time | $2.5144 \pm 0.0618$ | $2.7412 \pm 0.1238$ | $3.2003 \pm 0.1762$ | $3.3740 \pm 0.1817$ |
| | $\sigma$ | $0.9672 \pm 0.0041$ | $2.3931 \pm 0.0022$ | $2.7192 \pm 0.0022$ | $2.8144 \pm 0.0021$ |
| | $\rho$ | $1.2280 \pm 0.0030$ | $1.1158 \pm 0.0031$ | $0.8970 \pm 0.0030$ | $0.7875 \pm 0.0029$ |
| | $\beta$ | $0.2009 \pm 0.0010$ | $0.1735 \pm 0.0009$ | $0.2516 \pm 0.0009$ | $0.2960 \pm 0.0009$ |

In this section, we will show the experiments on the chaotic Lorenz system, one of the most well-known chaotic dynamical systems (Baines, 2008). The Lorenz system has three components

$(X, Y, Z)$ governed by the following ODEs, parameterized by $\beta$, $\rho$, and $\sigma$:

$$\frac{dX}{dt} = \sigma\,(Y - X),$$
$$\frac{dY}{dt} = X\,(\rho - Z) - Y, \tag{30}$$
$$\frac{dZ}{dt} = X\,Y - \beta\,Z.$$

We generated the ground-truth trajectory using initial conditions $(X(0), Y(0), Z(0)) = (5, 5, 5)$ and parameters $\theta = (\beta, \rho, \sigma) = \left(\frac{8}{3}, 28, 10\right)$, with 2561 time steps over the interval $T = [0, 40]$. We then selected 641 points from $T = [0, 20]$. As an illustrative example, we only consider discretizations with $161, 321, \ldots, 1281$ equally spaced time points. We simulated 100 datasets, and a Gaussian noise with standard deviation 0.1 across both components. The absolute errors for each parameter are summarized below. For the truncation numbers, we found that the performance will be stabilized at $l = 81$ and $j = 161$. From the Tab.S9, we can see that our EFiGP method consistently provides slightly better results for all parameters.

## S3.6 VISUALIZATION FOR EACH SYSTEM

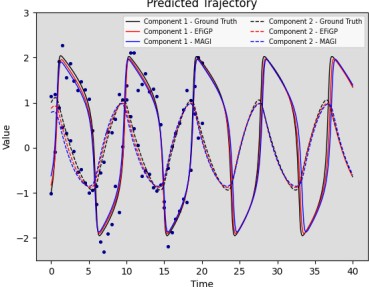

Figure S3: Predicted trajectory from EFiGP (red solid and dashed line) and from MAGI (blue solid and dashed line) for a 1281 discretization size on the FN system with ground-truth trajectory (black solid and dashed line) and 41 observed data points.

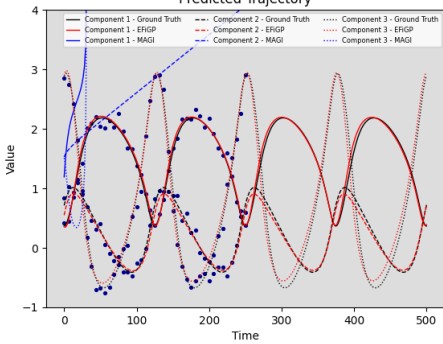

Figure S4: Predicted trajectory from EFiGP (red solid, dashed and dotted line) and from MAGI (blue solid, dashed and dotted line) for a 1281 discretization size on the log-transformed Hes1 system with ground-truth trajectory (black solid, dashed and dotted line) and 41 observed data points. At 1281 discretization, MAGI failed to converge while EFiGP still produce meaningful results.

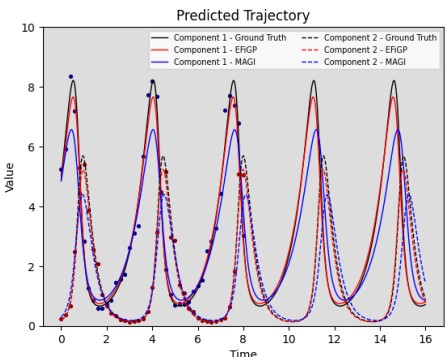

Figure S5: Predicted trajectory from EFiGP (red solid and dashed line) and from MAGI (blue solid and dashed line) for a 1281 discretization size on the LV system with ground-truth trajectory (black solid and dashed line) and 41 observed data points.

## S3.7 ERROR PLOT

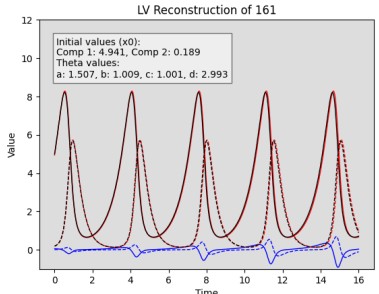 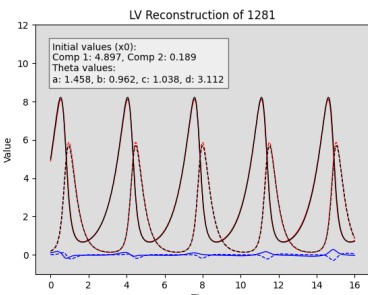

Figure S6: Single-dataset comparison between the predicted trajectory (red solid and dashed lines) and the ground-truth trajectory (black solid and dashed lines) at a discretization level of 161 (LHS) and 1,281 (RHS). Also shown is a different trajectory (blue solid and dashed lines) inferred using the estimated initial values and parameters. This comparison is conducted on a single dataset for illustrative purposes. The true parameter values are $a = 1.5$, $b = 1$, $c = 1$, and $d = 3$, with initial conditions $x_1(0) = 5$ and $x_2(0) = 0.2$. As evident from the figure, at a discretization level of 161 (LHS), the inferred parameters are more accurate. However, at a finer discretization level of 1,281 (RHS), the trajectory RMSE is lower despite greater parameter estimation errors. This highlights the phenomenon of weakly identifiable parameters, where a parameter set with higher error can still yield trajectories with improved accuracy.

