# OpenReview forum: "EFiGP: Eigen-Fourier Physics-Informed Gaussian Process for Inference of Dynamic Systems"
_ICLR.cc/2026/Conference — Submitted to ICLR 2026_

### Official Review · Reviewer_Jyix · 2025-10-22

**Soundness:** 3
**Presentation:** 2
**Contribution:** 2
**Rating:** 2
**Confidence:** 3

**Summary:**

The proposed Eigen-Fourier Physics-Informed Gaussian Process (EFiGP) framework demonstrates a certain degree of theoretical novelty by integrating Fourier transformation and eigen-decomposition into a physics-informed Gaussian process framework, aiming to enhance the efficiency and stability of ODE parameter inference and trajectory reconstruction.

**Strengths:**

1.Embedding the Fourier transform and eigen-decomposition into the GP framework is a non-trivial idea that strengthens the physical constraints from a frequency-domain perspective while reducing computational complexity.

2.The authors clearly identify the limitations of the MAGI framework—specifically, its high computational cost and poor convergence under dense discretization—and propose targeted solutions through frequency-domain and eigenspace dimensionality reduction. The problem statement is well articulated.

3.The experiments are conducted on three classical dynamical systems—FitzHugh-Nagumo, Lotka-Volterra, and Hes1—with clear comparative analyses. The results are quantitatively summarized in tables reporting RMSE and parameter errors, demonstrating that EFiGP achieves faster computation and more stable performance.

**Weaknesses:**

The paper mainly represents an engineering extension of the MAGI framework rather than a fundamental theoretical breakthrough. Although the idea of transferring ODE physical constraints into the Fourier domain is inspiring, its mathematical validity and convergence analysis are entirely absent. The paper provides no theoretical guarantees such as consistency, bias bounds, or truncation error analysis. Key derivations (e.g., Equations (9) and (11)) rely on statements like “by Lemma 2.2 we can obtain...” without rigorous justification of whether these transformations preserve the equivalence of the physical constraints. Moreover, there is no discussion of alignment with Bayesian inference theory (e.g., posterior consistency or marginal likelihood).

Although the proposed EFiGP framework shows computational advantages over MAGI, the experimental validation remains insufficient in both scope and depth.
All experiments are conducted on classical low-dimensional ODE benchmarks (FitzHugh–Nagumo, Lotka–Volterra, Hes1). These are toy systems with simple oscillatory dynamics, and the observation noise is i.i.d. Gaussian with uniform sampling—conditions that are relatively easy for GP-based methods. The paper does not test EFiGP on high-dimensional, chaotic, or real-world scientific datasets (e.g., climate models, fluid dynamics, or neural population dynamics), where scalability and robustness are critical. The comparison is restricted to MAGI and a simple numerical solver (Runge–Kutta). There are no experiments against more recent and competitive baselines such as FNO (Fourier Neural Operator, Li et al., ICLR 2021), AutoIP (Da Long et al., NeurIPS 2022), and Fenrir (Tronarp et al., ICML 2022), all of which address physics-informed kernel or operator learning in related contexts. The results mainly highlight faster computation, but accuracy improvements are marginal (Tables 2–5 show only small RMSE differences), and there is no systematic analysis of the trade-off between speed and inference accuracy as discretization increases.

The paper’s claims about computational complexity and scalability are also somewhat vague. The authors state that the complexity is reduced from O(n²) to a constant level (due to fixed j and l), but this is not theoretically rigorous—if j and l grow with system complexity, the computation may still scale linearly or quadratically. No detailed runtime environment or complexity–n scaling curves are provided; only static tables are shown.

The paper’s positioning and comparative analysis are also limited. It does not compare with recent works on physics-informed kernel learning (e.g., AutoIP) or FNO-style spectral operator methods. The Fourier domain truncation approach resembles the idea behind the Fourier Neural Operator, yet the paper does not clarify its relation or distinction from operator-learning frameworks.

There are also issues in writing and academic presentation. Some notations are redundant or ambiguous (e.g., W_F^I, m are not clearly defined).

**Questions:**

1. How does the time complexity of EFiGP scale with the state dimension D and the number of discretization points n? Is the method still feasible for large-scale systems?
2. How are the two truncation hyperparameters (l for Fourier, j for eigen) selected automatically? Could their choices lead to underfitting?
3. How does the method perform on non-periodic systems ?
4. Can you provide comparisons against Fenrir and AutoIP under identical settings?

---

> ### Author Response · Authors · 2025-11-14
> **Rebuttal Part 1**
>
> Thank you for your constructive feedback on this paper. We will address the concerns and questions you raised.
>
> - **Theoretical Analysis**: We appreciate that our idea is recognized as valuable, and we thank for pointing out the lack of theoretical analysis. Establishing a solid theoretical foundation for EFiGP, as well as for many physics-informed GP (PIGP) methods, remains an important but underexplored direction. Most existing approaches emphasize inference performance rather than formal guarantees, largely because they rely on the well-known **linearity of differential operators** and the **closure of GPs under linear transformations**. Although a full proof is beyond the current scope, **EFiGP is theoretically consistent with the standard GP framework**, as its Fourier and eigen decompositions are linear transformations that preserve the equivalence of the underlying physical constraints.
>
> - **Computational Complexity and Scalability**: Thank you for the insightful comments regarding computational cost. With respect to the discretization size $N$, you are absolutely correct that the computational cost of EFiGP increases linearly with both $j$ and $l$. More importantly, the final posterior is no longer dependent on the discretization itself; instead, it depends only on $j$ and $l$, both of which can be predetermined. We also sincerely thank for the valuable comment regarding scalability with the state dimension $D$. Since all components in our single-component formulation remain Gaussian, the full posterior for the log version is obtained by summing over all $D$ components. Consequently, the overall computational cost scales as $O(Dbl)$. Thus, EFiGP remains feasible for large-scale systems. To verify this, we conducted a set of experiments evaluating the computational cost across dimensions $D$, including the LV system $(D = 2)$, the Hes1 system $(D = 3)$, and the SEIR system $(D = 4)$, using $j = 81$, $l = 41$, and a discretization of 1281 points. The results are attached below for your reference. These empirical results are consistent with our theoretical $O(D)$ complexity. Although both MAGI and EFiGP exhibit approximately linear growth in runtime, EFiGP is substantially faster across all tested systems
>
> **Runtime Comparison**
> | Method / Setting  | **D = 4 (SEIR)** | **D = 3 (Hes1)** | **D = 2 (LV)** |
> | - | - | - | - |
> | **EFiGP** | 3.502 ± 0.068    | 2.990 ± 0.090| 2.100 ± 0.160  |
> | **MAGI**  | 20.823 ± 1.080   | 15.062 ± 0.429   | 9.714 ± 0.112  |
>
> - **FNO vs EFiGP**: You are absolutely correct that both FNO and EFiGP make use of Fourier transformations to improve performance. However, our method is fundamentally a **Gaussian Process based approach** for solving the *ODE inverse problem*, and to the best of our knowledge it is the first work specifically targeting this setting, rather than serving as a general ODE/PDE solver. In EFiGP, the Fourier transform is used to measure the discrepancy between the ODE dynamics and the GP representation in Fourier space (i.e., through $W_I^F$). In contrast, **Vanilla FNO is not physics informed**, and a **neural network–based** method that learns differential equations by parameterizing the integral kernel directly in Fourier space.
>
> - **Validation Insufficient**: Thank you for your suggestions regarding a more comprehensive comparison against benchmark methods such as AutoIP and Fenrir. Since AutoIP’s open-source code is not available, we now include a comparison with Fenrir for your reference. Due to space limitations, we present one experiment under a comparable setting: **1281 discretization** (discretization), **41 noisy observations** with noise scale **0.1**, on the LV system, averaged over 100 runs. The results are shown below, including the running time for the optimization step. From the table, we observe that both methods achieve comparable accuracy. Furthermore, the comparison between EFiGP and Fenrir shows that Fenrir yields better accuracy for parameters $a$, $c$, and $d$, while EFiGP yields better accuracy for parameter $b$. In our paper, we also conduct a sensitivity analysis on the initial guess of the ODE parameters, comparing our method with Fenrir and RK. We recognized that numerical–solver–based methods (RK and Fenrir) require a good initial guess to obtain meaningful results; otherwise, the optimization may diverge. From this experiment, we find that EFiGP is more user-friendly. Since numerical solvers are widely recognized as strong baselines, as suggested by [McGreivy and Hakim, 2024](https://www.nature.com/articles/s42256-024-00897-5), we choose the Runge–Kutta (RK) method due to its lower sensitivity.
>
>  **Table: Comparison between EFiGP and Fenrir**
>  | Method |   a   |  b |   c   |  d|    Time (s) |
>  |:---:|:----:|:--:|:--:|:--:|:--:|
>  | **EFiGP**| 0.0780 ± 0.0240| 0.0400 ± 0.0260| 0.0730 ± 0.0190|0.1590 ± 0.0340| 1.5670 ± 0.0388|
>  | **Fenrir** | 0.0573 ± 0.0413| 0.0430 ± 0.0270| 0.0312 ± 0.0235|0.1004 ± 0.0726| 2.7431 ± 0.0755|

---

> ### Author Response · Authors · 2025-11-14
> **Rebuttal Part 2**
>
> We apologize for splitting the rebuttal into two sections due to the character limits. Let us continue to address your questions and concerns.
>
> - **More on validation**: Thank you for your suggestions regarding chaotic systems. **We acknowledge that our method provides limited gains in accuracy for non-oscillatory ODEs**. To further illustrate performance, we include two additional examples. The first is the chaotic Lorenz system. We generated the ground-truth trajectory using initial conditions $(X(0), Y(0), Z(0)) = (5, 5, 5)$ and parameters $\theta = (\beta, \rho, \sigma) = \left(\tfrac{8}{3}, 28, 10\right),$ with 2561 discretization over the interval $T = [0, 40].$ We then selected 641 points from the sub-interval $T = [0, 20]$ and added Gaussian noise with standard deviation 0.1. The absolute parameter-estimation errors over 100 runs are summarized below. Following our tuning procedure, we used truncation numbers $l = 81, j = 161,$ and both EFiGP and MAGI were run with a discretization of 1281 points. From this table, we observe that EFiGP provides slightly better estimates for $\gamma$ and $\beta$. Due to space limitations, the results for additional discretization settings are provided in Appendix S3.5 (Table S9).
>
>  **Table: Comparison on the chaotic Lorenz system**
>  | Method | $\gamma$ (mean ± SD)      | $\rho$ (mean ± SD)      | $\beta$ (mean ± SD)|
>  |--------|---------------------|---------------------|---------------------|
>  | MAGI   | 2.8842 ± 0.0004     | 0.3378 ± 0.0026     | 0.3984 ± 0.0007     |
>  | EFiGP  | 2.8144 ± 0.0021     | 0.7875 ± 0.0029     | 0.2960 ± 0.0009     |
>
> Furthermore, we include an additional example **on the SEIR system** for your reference. We generated the ground-truth trajectory using initial conditions $(S(0), E(0), I(0), R(0)) = (0.99, 0.01, 0, 0)$ and parameters $\theta = (\beta, \sigma, \gamma) = (1, 1, 0.1),$ with 2561 discretization points over the interval $T = [0, 20].$ We then selected 41 points from the sub-interval $T = [0, 10]$ and added Gaussian noise with standard deviation 0.05. Both MAGI and EFiGP were run with a discretization of 1281 steps, $l = 81, j = 41,$ and results were averaged over 20 runs. The absolute estimation errors for each parameter and the trajectory RMSE over the 2561-point discretization are summarized in the table below. From this table, we observe that EFiGP consistently provides slightly better estimates across all parameters and trajectory RMSEs. More importantly, the performance of EFiGP for component **E** is nearly twice as accurate as that of MAGI. Compared with the RK method, EFiGP also provides comparable results, especially for $\sigma$, $\gamma$, component **I**, and component **R**. Based on these two non-oscillatory ODE examples, we acknowledge the current limitations of our method and plan to address them in future work.
>
>  **Table: Comparison on the SEIR system**
>  | Method | $\beta$   | $\sigma$   |  $\gamma$  | Component S | Component E | Component I | Component R |
>  |--------|-------------|-------------|-------------|-------------|-------------|-------------|-------------|
>  | EFiGP  | 0.272 ± 0.130 | 0.172 ± 0.080 | 0.043 ± 0.021 | 0.0464 ± 0.0135 | 0.0161 ± 0.0066 | 0.0924 ± 0.0469 | 0.1006 ± 0.0491 |
>  | MAGI   | 0.276 ± 0.122 | 0.365 ± 0.153 | 0.046 ± 0.026 | 0.0456 ± 0.0256 | 0.0385 ± 0.0250 | 0.1038 ± 0.0535 | 0.1181 ± 0.0659 |
>  |RK | 0.0622 ± 0.0000 | 0.2137 ± 0.0000 | 0.0955 ± 0.0000 | 0.0134 ± 0.0000 | 0.0144 ± 0.0000 | 0.1421 ± 0.0000 | 0.1383 ± 0.0000 |
>
>
> - **Tuning of Truncation Numbers**: Thank you for your insightful comment regarding the truncation hyperparameters. In our method, we selected the truncation numbers by **gradually increasing them from lower levels** (e.g., $l = 11, j = 11$) until the results stabilized, as shown in Figures S1 and S2 in the Appendix. From these heatmaps, we observe that the performance stabilizes as both $j$ and $l$ increase. Thus, further increases in the truncation numbers do not yield meaningful improvements. This indicates that once the truncation parameters reach this stabilized regime, the model is not underfitted. We apologize for not presenting the full tuning process in the main text due to page limits. A comprehensive description of the truncation-selection procedure is now included in Appendix S.3.4.
>
> - **Academic Presentation**: Thank you for pointing out the ambiguous notation. We will refine the notation in the final version of the paper.
>     - For example, $W_I^{\mathcal{F}} = \\{ \mathcal{F}[\dot X(t)] - \mathcal{F}[f(X(t),\theta,t)] \\}_{t \in I}$
>     - $m(I)= {'\mathcal{K}}(I,I)\mathcal{K}(I,I)^{-1}$, where $'\mathcal{K}(s,t) = \frac{\partial}{\partial{s}}\mathcal{K}(s,t)$.

---

> > ### Comment · Reviewer_Jyix · 2025-11-22
> >
> > Thank you for the detailed rebuttal. Since the AutoIP implementation released by the original authors is **publicly available on GitHub**, a comparison is essential to properly contextualize the contribution. AutoIP (Long et al., ICML 2022) addresses the same inverse problem and is a highly relevant baseline; without this experiment, it is difficult to fully assess the contribution relative to state-of-the-art.
> >
> > Regarding the Fenrir comparison, the added experiment is a positive development, but the current evaluation remains too limited to support strong conclusions.. Testing only a single setting is insufficient for drawing robust conclusions. Under such a limited protocol, EFiGP’s purported advantages over this 2022 baseline are not convincingly demonstrated (e.g., some parameter errors are quite large), and the numerical superiority reported so far requires validation across a broader and more representative range of discretizations, observation counts, and noise levels.
> >
> > I also want to restate my earlier comment on FNO more clearly. My point was not about “vanilla FNO,” but about the substantial line of FNO-style operator learning methods explicitly developed for inverse problems. This is now an active and fast-growing literature, including Fourier-based inverse solvers (arXiv:2402.11722; arXiv:2505.08740) and operator inverse design approaches (arXiv:2301.11167), all of which leverage Fourier/spectral structures in ways directly relevant to your setting. In that context, the statement that “vanilla FNO is not physics-informed” reads as overly narrow and risks misrepresenting the broader operator-learning landscape. At minimum, this related literature should be discussed accurately, and where feasible, representative comparisons should be included.
> >
> > Finally, as an engineering extension of the MAGI framework, comparing only with MAGI (2021) is not sufficient to support a 2026 submission. Unless the paper provides substantial theoretical guarantees—which it does not appear to—its empirical results need to be compared with more recent and well-established baselines from 2022–2024 in order to properly frame the method and validate its claimed improvements
> >
> > I appreciate the authors' thoughtful responses to my comments, the additional experiments, and the relevant feedback from other reviewers. However, several critical gaps remain that significantly limit the strength of the current submission. The experimental design, while improved, does not yet provide sufficiently robust evidence to support the key claims regarding the method's advantages over recent state-of-the-art approaches. More comprehensive comparisons and broader experimental validation are needed before the reported improvements can be considered convincingly demonstrated.

---

> > > ### Author Response · Authors · 2025-11-28
> > > **AutoIP comparison (part 1 of 3 )**
> > >
> > > Thank you for your constructive feedback on this paper again. We will address the concerns and questions you raised.
> > >
> > > - **Compared with more recent and well-established baselines:** You are absolutely correct that comparing against more recent baselines will make our paper more convincing. Therefore, we will compare our method with **Fenrir** and **AutoIP**
> > >
> > > - **Comparsion with AutoIP:** We also thank the reviewer for noting that the AutoIP code is publicly available. Upon examining the repository, we found that the provided implementation is designed as an illustrative demonstration of their methodology and is specialized to the specific ODE/PDE systems considered in their paper. Adapting AutoIP to our experimental settings would require substantial re-implementation effort due to technical details and time constraints. Despite these challenges, we conducted comparisons with both **AutoIP** and **Fenrir** under the **damped and undamped ODE** setting.
> > >
> > >      We follow their data-generating process by using **SciPy** with the initial position set to $3/4\pi$, an initial velocity of 0, and a time interval from $t = 0$ to $t = 28.8$ with a step size of 0.01 for undamped ODE. Also, we use $b=0.2$ for damped ODE as well. After generating the data, we adhere to our experimental setup by varying the number of observations (i.e., 41, 81, 161, and 321), using a noise scale of 0.1 for both system (i.e., damped and undamped). For EFiGP, we fix the discretization at 1281 and use truncation parameters (j = 81) and (l = 41). For Fenrir and AutoIP, we directly use the code provided by the authors. We report the reconstructed RMSE for both methods under both the damped and undamped cases. The results are attached below for your reference.
> > >
> > >     From this table, we observe that the results obtained by EFiGP and Fenrir are consistently better than those of AutoIP across all cases and for both systems. When comparing EFiGP with Fenrir, we see that the performance of both methods improves as more observations are provided, and their accuracies are generally comparable. For example, in some settings Fenrir achieves an RMSE only about 0.0002 lower than EFiGP. Therefore, we believe that EFiGP can provide results at a similar precision level to Fenrir. Regarding AutoIP, we note that its performance in our experiments is on the same scale as the results reported in its original paper, indicating that our evaluation of AutoIP is consistent with the authors’ findings. As shown in their paper, their method focuses on training a Gaussian process for the ODE function. We think their approach is more compatible with purely Gaussian process.
> > >
> > > **Table: Comparison result of damped and undamped ODE**
> > > | **System Type** | **Method** | **41**          | **81**          | **161**         | **321**         |
> > > | --- | -- | --- | --- | ---- | -- |
> > > | **No Damping**| EFIGP | 0.2009±0.1249 | 0.1052±0.0719 | 0.0952±0.0646 | 0.0696±0.0324 |
> > > || Fenrir   |  0.1937±0.1120 | 0.1050±0.0807 | 0.0751±0.0628 | 0.0540±0.0323 |
> > > || AutoIP| 0.4723±0.0979 | 0.5046±0.0652 | 0.5058±0.0876 | 0.5084±0.1011 |
> > > | **Damping**|  EFIGP | 0.0188±0.0109 | 0.0104±0.0042 | 0.0087±0.0036 | 0.0042±0.0017 |
> > > || Fenrir  | 0.0129±0.0087 | 0.0097±0.0054 | 0.0062±0.0037 | 0.0039±0.0013 |
> > > ||   AutoIP | 0.0923±0.0122 | 0.1459±0.0140 | 0.1659±0.0080 | 0.1833±0.0167 |

---

> > > ### Author Response · Authors · 2025-11-28
> > > **Fenrir comparison (part 2 of 3 )**
> > >
> > > - **Comparsion with Fenrir and numerical solver:**  You are absolutely correct that comparing our method to Fenrir under only a single experimental setting is insufficient. We then conducted a more comprehensive comparison by **varying the number of observations** on both the **FN** and **LV** systems. We also included the RK method as an additional baseline, as it is widely regarded as a gold-standard numerical solver. Due to convergence issues encountered with the JAX optimizer, we were unable to perform an evaluation on the Hes1 system for NM and Fenrir. We used a noise level of 0.2 for the FN system and a log-normal noise level of 0.1 for the LV system. The ODE setup follows exactly the same configuration as in our main paper. For EFiGP, the discretization size was fixed at 1281 across all experiments to ensure consistency. The results are generated by 20 repetitions for each case.
> > >
> > >    From the FN results, we observe that the performance of the inferred trajectories consistently improves as the number of observations increases across all methods. With only 41 observations, the RK and Fenrir methods achieve RMSE values on $x_1$ that are roughly half of those obtained by EFiGP. However, when the number of observations increases to 321, the inferred trajectories produced by EFiGP become comparable to those of the RK method. More importantly, for the inferred parameter $c$, EFiGP yields more accurate estimates than both RK and Fenrir. A similar pattern appears in the LV experiments. As the number of observations increases, the quality of the inferred trajectories improves for all methods. Even with 41 observations, the recovered trajectory of the first component obtained by EFiGP is already comparable to those produced by the RK and Fenrir methods. Taking into account the sensitivity of the NM and Fenrir methods to the choice of initial guesses (as demonstrated in Appendix S3.2), we conclude that EFiGP achieves a comparable level of precision while being substantially easier to use. Particularly for users who may not possess expert knowledge needed to tune initial conditions for NM or Fenrir.
> > >
> > > **Table: Comparison result of FN system**
> > > | **Method**   | **Param**   | **41 obs**| **81 obs**| **161 obs**| **321 obs**|
> > > |- |---|--|---|--|--|
> > > | **EFiGP** | a       | 0.031±0.024| 0.027±0.016| 0.015±0.008    | 0.017±0.006    |
> > > |         | b       | 0.233±0.103| 0.126±0.019| 0.099±0.025    | 0.086±0.016    |
> > > |         | c       | 0.050±0.034| 0.027±0.014| 0.024±0.008    | 0.019±0.008    |
> > > |         | $x_1$     | 0.276±0.123| 0.122±0.042| 0.104±0.041    | 0.077±0.014    |
> > > |         | $x_2$      | 0.091±0.038| 0.041±0.008| 0.037±0.007    | 0.034±0.010    |
> > > | **Fenrir**| a | 0.016±0.017| 0.019±0.016| 0.012±0.010    | 0.011±0.007    |
> > > |           | b | 0.114±0.045| 0.070±0.050| 0.037±0.019    | 0.039±0.025    |
> > > |           | c | 0.057±0.037| 0.036±0.019| 0.017±0.011    | 0.023±0.013    |
> > > |           | $x_1$  | 0.140±0.069| 0.094±0.056| 0.066±0.021    | 0.051±0.022    |
> > > |           | $x_2$   | 0.057±0.022 | 0.038±0.018    | 0.021±0.008    | 0.020±0.012    |
> > > | **RK**  |a       | 0.041±0.023    | 0.019±0.017| 0.022±0.015    | 0.016±0.011    |
> > > |         | b       | 0.126±0.010 | 0.109±0.016| 0.051±0.040    | 0.047±0.031    |
> > > |         | c       | 0.038±0.022| 0.021±0.017| 0.023±0.020    | 0.026±0.014    |
> > > |         | $x_1$       | 0.127±0.076| 0.084±0.034| 0.062±0.035    | 0.057±0.025    |
> > > |         | $x_2$      | 0.064±0.029| 0.033±0.008    | 0.022±0.004    | 0.020±0.004    |
> > >
> > > **Table: The comparision result of LV system**
> > > | **Method**   | **Param**   | **41**        | **81** | **161**       | **321**        |
> > > |--|--|--|--|--|--|
> > > | **EFiGP** | a | 0.078±0.024| 0.094±0.012| 0.055±0.009 | 0.034±0.005 |
> > > |           | b | 0.040±0.026 | 0.047±0.036| 0.041±0.024 | 0.029±0.024 |
> > > |           | c | 0.073±0.019 | 0.070±0.009 | 0.046±0.008 | 0.032±0.002 |
> > > |           | d | 0.159±0.034 | 0.185±0.038 | 0.097±0.025 | 0.075±0.008 |
> > > |           | log($x_1$) | 0.064±0.019 | 0.056±0.011 | 0.043±0.009 | 0.038±0.002 |
> > > |           | log($x_2$) | 0.058±0.024 | 0.049±0.011 | 0.048±0.016 | 0.041±0.002 |
> > > | **Fenrir** | a  | 0.057 ± 0.041 | 0.030±0.022| 0.028±0.022     | 0.027±0.018     |
> > > | | b          | 0.043 ± 0.027 | 0.031±0.031 | 0.016±0.013| 0.014±0.009|
> > > | | c          | 0.031 ± 0.023 | 0.016±0.013| 0.013±0.014| 0.010±0.011 |
> > > | | d          | 0.100 ± 0.073  | 0.094±0.045| 0.058±0.034| 0.064±0.030|
> > > | |log($x_1$)   | 0.066±0.034| 0.043±0.021| 0.041±0.029 | 0.031±0.016 |
> > > |  |log($x_2$)   | 0.048±0.029| 0.044±0.023| 0.034±0.022 | 0.023±0.011|
> > > | **RK** | a | 0.460±0.046 | 0.457±0.037 | 0.470±0.014 | 0.437±0.010 |
> > > | | b | 0.021±0.011 | 0.050±0.029 | 0.041±0.009 | 0.030±0.012 |
> > > | | c | 0.018±0.022 | 0.020±0.014 | 0.015±0.007 | 0.013±0.008 |
> > > | | d | 0.090±0.036 | 0.081±0.064 | 0.048±0.023 | 0.061±0.016 |
> > > | | log($x_1$)| 0.056±0.023 | 0.043±0.017 | 0.034±0.009 | 0.032±0.007 |
> > > | | log($x_2$)| 0.034±0.008 | 0.026±0.006 | 0.023±0.008 | 0.020±0.004 |

---

> ### Author Response · Authors · 2025-11-28
> **Discussion on FNO (part 3 of 3)**
>
> - **More discussion about FNO:** You are absolutely correct that the statement “vanilla FNO is not physics-informed” reads as overly narrow. We apologize for the previous wording and would like to refine it below.
>
>     The classical FNO is designed to learn a solution operator that maps input functions, such as initial or boundary conditions (and possibly coefficient fields), to the corresponding solution (i.e., the full trajectory). Therefore, if we directly apply an FNO to observational data together with the initial condition, it will approximate the forward solution map and output a predicted trajectory. In contrast, our method is designed within a Bayesian framework. We sincerely thank you for directing us to these papers. However, [1] and [2] focus primarily on PDE inverse problems. Although [3] is designed to handle ODEs and the GitHub repository is publicly listed, the corresponding **code and usage instructions have not yet been released**. Thus, at the current stage, we will include a discussion of operator-learning methods, as numerous recent works have extended them to inverse problems. Furthermore, we will include this discussion in our literature review section as a new paragraph following the second paragraph.
>
>     *For solving the invserse problem, neural operator (NO) learning methods are extended for the inverse problem. [3] propose a neural inverse operator (NIO) specifically designed to handle inverse problems by integrating DeepONet and FNO in a sequential manner. However, NIO does not offer uncertainty quantification. To enable the uncertainty quantification, [1] introduces an invertible Fourier Neural Operator (IFNO) designed to jointly address forward and inverse problems by using the variational auto-encoder and an invertible Fourier blocks. More recently, [2] introduces the sensitivity-based loss terms to Neural Operators by incorporating a sensitivity loss to enhance their predictive accuracy and performance, which is constructred using Jacobian of the predicted outputs with respect to the input parameters.*
>
>
> [1] Long et al, Invertible Fourier Neural Operators for Tackling Both Forward and Inverse Problems, AISTAT, 2025.
>
> [2] Behroozi et al, Sensitivity-Constrained Fourier Neural Operators for Forward and Inverse Problems in Parametric Differential Equations, ICLR, 2025.
>
> [3] Molinaro et al, Neural Inverse Operators for Solving PDE Inverse Problems, ICML, 2023.
>
> We sincerely thank you for your thoughtful comments and valuable suggestions. We hope that our responses have fully addressed your concerns.

---

> > ### Comment · Reviewer_Jyix · 2025-11-28
> >
> > I appreciate the authors’ substantial efforts during the rebuttal period, including the comparison with AutoIP, the extended Fenrir experiments across multiple observation settings, and the improved discussion of FNO-related literature. These additions are valuable and help address several of my earlier questions.
> >
> > However, to further strengthen the quality and clarity of this submission, I still retain the following reservations:
> >
> >
> >
> > **1. Limited accuracy improvements over existing baselines:** In the comprehensive comparisons on the FN and LV systems, Fenrir (2022) outperforms EFiGP on the majority of parameter estimates and trajectory RMSEs. The authors' primary justification is "ease of use" (robustness to initial guesses), but this alone does not constitute a substantial improvement over the state-of-the-art. If a practitioner is willing to invest effort in tuning initialization, Fenrir appears to be the stronger choice.
> >
> > **2. Incomplete efficiency data:** While the initial rebuttal indicated that EFiGP is approximately 1.7× faster than Fenrir (1.57s vs 2.74s), the subsequent extended experiments do not include runtime comparisons. To substantiate the claimed computational advantages, the authors should provide comprehensive timing data across all experimental settings—not only against MAGI but also against Fenrir and other baselines.
> >
> > **3. Unclear evaluation of overall parameter estimation quality:** For ODE parameter estimation, it remains unclear how to assess the overall quality when different methods perform better on different parameters. The current comparison lacks a principled way to weigh the importance of individual parameters—in practice, some parameters may have a much greater impact on system dynamics than others. A more meaningful evaluation might consider parameter sensitivity analysis or downstream task performance rather than treating all parameter errors equally.
> >
> > Overall, while the paper presents a useful engineering contribution, the above concerns prevent me from fully endorsing the submission in its current form

---

> > > ### Author Response · Authors · 2025-11-29
> > >
> > > Thanks for your positive and constructive feedback. We are glad to know that our response has addressed most of your questions.
> > >
> > > - **Accuracy Improvements:** Thank you for your additional comments regarding Fenrir. Fenrir is a probabilistic numerical method, essentially a **numerical solver with extra probabilistic functionality**. Therefore, as discussed in [Kersting et al, 2020](https://proceedings.mlr.press/v119/kersting20a/kersting20a.pdf), it fundamentally differs from our proposed approach. Since numerical methods are widely recognized as strong baselines and the **gold standard**, as suggested by [McGreivy and Hakim, 2024](https://www.nature.com/articles/s42256-024-00897-5), we include the RK method to illustrate how close our performance is to this standard. Therefore, we would like to clarify that Fenrir serves **as another gold standard method** rather than a competing approach. Beyond purely numerical comparisons, we additionally provide an initial guess sensitivity analysis for our method, RK, and Fenrir (Appendix S3.2). These results indicate that our method is easy to apply and does not rely on precise prior knowledge of the system.
> > >
> > >
> > > - **Comoutational Cost:** You are absolutely correct that including a computational cost table for the new benchmark can strengthen the evaluation of our method’s efficiency. We have therefore added the computational cost results for Fenrir on the FN and LV systems, as well as AutoIP and Fenrir on the damped and non-damped systems, for your reference. From the following two tables, we observe that EFiGP is approximately **1.5–1.7× faster** than Fenrir across all systems and cases. Compared with the RK method, EFiGP is roughly **10× faster**. As we stated earlier, AutoIP is more compatible with a purely Gaussian process framework, since it focuses on training a Gaussian process to approximate the ODE function. Therefore, overall, **EFiGP is the most efficient method** among the compared approaches. Combined with its insensitivity to the initial guess, we conclude that **EFiGP is user-friendly and can deliver results with comparable precision while saving computational cost**.
> > >
> > > **Time Comparison Table (damped and undamped ODE)**
> > >
> > > | Setting        | Method     | 41 obs   | 81 obs           | 161 obs             | 321 obs            |
> > > | -------------- | ---------- | --------------- | --------------- | --------------- | --------------- |
> > > | **Damping**    | **EFiGP**  | 1.4155 ± 0.0192 | 1.4288 ± 0.0180 | 1.4215 ± 0.0337 | 1.4336 ± 0.0219 |
> > > |                | **Fenrir** | 2.9213 ± 0.0463 | 2.7158 ± 0.0665 | 2.7304 ± 0.0463 | 2.7794 ± 0.0592 |
> > > |                | **AutoIP** | 27.6863 ± 0.2151  | 41.1907 ± 0.3108|  76.2882 ± 0.6913 | 181.6150 ± 22.0063 |
> > > | **No Damping** | **EFiGP**  | 1.4309 ± 0.0444 | 1.4185 ± 0.0149 | 1.4401 ± 0.0215 | 1.4350 ± 0.0200 |
> > > |                | **Fenrir** | 2.9480 ± 0.0544 | 2.9137 ± 0.0621 | 2.8639 ± 0.0698 | 2.7946 ± 0.0588 |
> > > |                | **AutoIP** | 30.6482 ± 4.0313 | 43.0049 ± 1.3814 |  77.5054 ± 1.7681 | 184.6488 ± 22.9703|
> > >
> > >
> > > **Time Comparison Table (FN & LV Systems)**
> > > | System | Method | 41 obs       | 81 obs       | 161 obs      | 321 obs      |
> > > | ------ | ------ | ------------ | ------------ | ------------ | ------------ |
> > > |  **FN** | EFiGP  | 1.84 ± 0.01  | 1.84 ± 0.02  | 1.96 ± 0.08  | 2.00 ± 0.05  |
> > > | | RK     | 11.37 ± 0.41 | 11.47 ± 0.26 | 11.35 ± 0.41 | 11.28 ± 0.57 |
> > > | | Fenrir | 2.80 ± 0.06  | 3.20 ± 0.06  | 3.15 ± 0.05  | 2.97 ± 0.05  |
> > > |  **LV** | EFiGP  | 1.58 ± 0.01  | 2.09 ± 0.06  | 2.10 ± 0.05  | 2.13 ± 0.08  |
> > > | | RK     | 11.24 ± 0.61 | 10.64 ± 0.59 | 11.03 ± 0.11 | 11.49 ± 0.11 |
> > > | | Fenrir | 2.74 ± 0.08  | 3.01 ± 0.05  | 2.98 ± 0.05  | 2.91 ± 0.07  |
> > >
> > > - **Evaluation of overall parameter estimation quality:** Thank you for your insightful comment regarding the importance of overall quality and more meaningful evaluation metrics. To the best of our knowledge, the standard approach for assessing parameter inference is to report the absolute error for each parameter. Developing new, more comprehensive metrics is beyond the scope of the current study. However, we agree that this represents an interesting and valuable direction for future research.
> > >
> > > We hope that our clarifications and additional experiments satisfactorily address your concerns, and we appreciate your constructive feedback in helping strengthen this work.

---

### Official Review · Reviewer_99Ym · 2025-10-22

**Soundness:** 2
**Presentation:** 3
**Contribution:** 2
**Rating:** 4
**Confidence:** 3

**Summary:**

The paper proposes an algorithm called EFiGP (Eigen-Fourier Physics-Informed Gaussian Process), which is a Bayesian framework for parameters estimation and trajectory reconstruction of data-driven dynamical systems governed by ordinary differential equations (ODEs). The algorithm is mainly built on MAGI (Manifold-Constrained Gaussian Process Inference) but brings truncation with Fourier Transformation and Eigen-decomposition so that the computation time and cost are reduced, while computational efficiency and accuracy are enhanced. Simulations are done on three datasets.

**Strengths:**

- EFiGP’s most significant strength lies in its ability to bypass numerical integration during parameter estimation and trajectory inference without repeated ODE solving.
 - The incorporation of eigen-decomposition and Fourier-domain truncation within a physics-informed Gaussian Process (GP) framework reduces computational complexity, resulting in near-constant runtime across increasing discretization levels. The proposed spectral and eigen-based reparameterization reduces computational complexity from O(n^2) to near-constant runtime with respect to discretization size, outperforming existing approaches such as MAGI.
 - Moreover, EFiGP maintains convergence under dense discretization settings where the baseline MAGI algorithm fails to converge.

**Weaknesses:**

- Despite its efficiency gains, the claimed improvements in estimation accuracy over baseline MAGI remain modest. In particular, while EFiGP slightly outperforms MAGI for dense discretization (≥161 points), MAGI has comparable accuracy for sparser datasets, shown in Table 2 and Table 3, where EFiGP’s trajectory estimation accuracy improves only when the discretization becomes sufficiently dense. The algorithm did not deal with the challenge posed by data sparsity. The approach remains sensitive to data sparsity, and the paper’s results suggest that additional observations are required to mitigate degradation in parameter estimation performance.
 - Furthermore, parameter identifiability issues persist, especially for weakly identifiable parameters such as b in the FN system. Although EFiGP improves over MAGI in this respect, the mean of absolute error for b (0.176 at 41-point discretization) remains substantial compared to the true value of 0.2, and the authors do not propose any strategies to address this limitation.
 - The main evaluations are on three oscillatory systems (FN, LV, Hes1), with only a brief mention of a chaotic system in the supplement. Since the algorithm’s efficiency relies on Fourier-domain representations that are naturally suited to oscillatory behavior, broader testing on non-oscillatory or chaotic systems would be essential to substantiate claims of general applicability and robustness.

**Questions:**

For the reduce in computation cost, how much comes from eigen-decomposition and how much comes from Fourier truncation?

---

> ### Author Response · Authors · 2025-11-14
>
> Thanks for your valuable time and insightful comments on our paper! We will address the concerns and questions you raised.
>
> - **Non-oscillatory ODE** You are absolutely correct that broader testing on non-oscillatory systems would be essential. We now include an additional **example on the SEIR system** for your reference. We generated the ground-truth trajectory using initial conditions $(S(0), E(0), I(0), R(0)) = (0.99, 0.01, 0, 0)$ and parameters $\theta = (\beta, \sigma, \gamma) = (1, 1, 0.1), $ with 2561 discretization points over the interval $T = [0, 20].$ We then selected 41 points from the sub-interval $T = [0, 10]$ and added Gaussian noise with standard deviation 0.05. Both MAGI and EFiGP were run with a discretization of 1281 steps, $l = 81, j = 41,$ and results were averaged over 20 runs. The absolute errors for each parameter and the trajectory RMSE over the 2561 discretization are summarized in the table below. From this table, we observe that EFiGP consistently provides slightly better estimates across all parameters and trajectory RMSEs. More importantly, the performance of EFiGP for component **E** is nearly twice as accurate as that of MAGI. Compared with the RK method, EFiGP also provides comparable results, especially for $\sigma$, $\gamma$, component **I**, and component **R**. Based on the SEIR example and the chaotic Lorenz system as shown in the Appednix S3.5. We acknowledge that our method provides **limited gains in accuracy** for non-oscillatory ODEs, and we will address them in future work. However, the **computational cost savings are still beneficial** for non-oscillatory ODEs.
>
> **Table: Comparison on the SEIR system**
> | Method | $\beta$   | $\sigma$   |  $\gamma$  | Component S | Component E | Component I | Component R |
> |--|--|--|--|--|--|--|--|
> | EFiGP  | 0.272 ± 0.130| 0.172 ± 0.080| 0.043 ± 0.021|0.0464 ± 0.0135|0.0161 ± 0.0066| 0.0924 ± 0.0469 | 0.1006 ± 0.0491 |
> | MAGI   | 0.276 ± 0.122| 0.365 ± 0.153| 0.046 ± 0.026|0.0456 ± 0.0256|0.0385 ± 0.0250| 0.1038 ± 0.0535 | 0.1181 ± 0.0659 |
> |RK | 0.0622 ± 0.0000| 0.2137 ± 0.0000| 0.0955 ± 0.0000| 0.0134 ± 0.0000 | 0.0144 ± 0.0000 | 0.1421 ± 0.0000 | 0.1383 ± 0.0000 |
>
> - **Computation Cost**: You are absolutely correct that showing the computational cost by separating the eigen and Fourier components makes the comparison clearer. We conducted a set of experiments evaluating the computational cost under three settings: **Full Eigen** (i.e., fixing the Fourier truncation at its optimal level), **Full Fourier** (i.e., fixing the eigen truncation at its optimal level), and **Optimal** (i.e., both eigen and Fourier truncations set to their optimal levels). More importantly, we examined the computational cost across different systems, including the LV system, the Hes1 system, and the SEIR system, using $j = 81$, $l = 41$, and a discretization of 1281 points. The results are attached below for your reference. From the table, we can see that both the eigen and Fourier components play important roles, but the eigen truncation provides slightly greater benefits. Fortunately, these empirical results also reveal the scalability of our method with respect to the ODE dimension $D$. Although both MAGI and EFiGP exhibit approximately linear growth in runtime, EFiGP is substantially faster across all tested systems. Note that the current computational costs are based on experiments run on a MacBook with an M3 chip.
>
> **Runtime Comparison**
>
> | Method / Setting         | **D = 4 (SEIR)** | **D = 3 (Hes1)** | **D = 2 (LV)** |
> | -- | -- | -- | -- |
> | **EFiGP – Full eigen**| 7.135 ± 0.170| 5.545 ± 0.244| 3.920 ± 0.190|
> | **EFiGP – Full Fourier**| 6.969 ± 0.525| 4.481 ± 0.122| 3.560 ± 0.100|
> | **EFiGP – Optimal**| 3.502 ± 0.068| 2.990 ± 0.090| 2.100 ± 0.160|
> | **MAGI**| 20.823 ± 1.080|15.062 ± 0.429| 9.714 ± 0.112|
>
> - **Dense discretization and Performance degradation** You are absolutely correct that using a denser discretization would improve performance. We would like to clarify that EFiGP is a collocation-based method, similar in spirit to finite element or spectral approaches, where dense discretization is a standard and necessary setting to accurately capture system dynamics. The goal of EFiGP is not to handle extremely sparse observations but to achieve efficient and stable inference under dense discretization, where conventional MAGI becomes computationally prohibitive. In this regime, EFiGP consistently demonstrates superior accuracy and significantly reduced computational cost. Besides the concern about discretization, regarding the deterioration in accuracy, to the best of our knowledge this behavior is largely due to the weak identifiability inherent in many ODE systems. In the paper, we provide empirical evidence showing that increasing the number of observations can help mitigate this issue. Technically, although a solution is beyond the current scope of our paper, we acknowledge that it is an important direction for future research.

---

> > ### Comment · Reviewer_99Ym · 2025-11-22
> >
> > I thank the reviewers for the clarifications. My questions are addressed.
> >
> > The last part your reply brought up connections to
> > Harkonen et al, Gaussian Process Priors for Systems of Linear Partial Differential Equations with Constant Coefficients, ICML 2023
> > in my mind. Can you perhaps also comment on this paper as well? A comparison is probably not suitable, as this paper addresses PDEs.
> >
> > Before changing my opinion, I would like to see an answer of the last reviewer, whether their comments are addressed as well.

---

> > > ### Author Response · Authors · 2025-11-28
> > >
> > > Thank you for letting us know that our response has addressed your question.
> > >
> > > - **Comment on the paper:** Thank you for pointing out this fantastic method to us. We plan to include it in our literature review section and will place it at the end of the first paragraph.
> > >
> > >     *More imporatnly, [1] uses the Ehrenpreis–Palamodov fundamental principle to embed a linear PDE (or system of PDEs) into the prior of a Gaussian process. This enables a kernel that ensures all sample paths satisfy the PDE, which is a powerful, mathematically rigorous way to inject PDE structure into a GP.*
> > >
> > > [1] Harkonen et al, Gaussian Process Priors for Systems of Linear Partial Differential Equations with Constant Coefficients, ICML 2023.
> > >
> > >
> > > - **More experiemnts:** For your interest, as suggested by the last reviewer, we have also provided two new sets of benchmarks (**AutoIP and Fenrir**) across four dynamical systems by varying the number of observations, using the identical experimental settings as in our comparison with the NM method (Appendix S3.1). Specifically, we evaluate AutoIP and Fenrir together with our method on the nonlinear pendulum system (with and without damping), and we additionally evaluate Fenrir on both the LV and FN systems.
> > >
> > > Thank you again for your valuable time and feedback.

---

### Official Review · Reviewer_ev1R · 2025-10-26

**Soundness:** 3
**Presentation:** 2
**Contribution:** 2
**Rating:** 6
**Confidence:** 4

**Summary:**

This paper is about using Gaussian processes (GPs) for calibrating dynamical systems governed by ODEs in a Bayesian manner. Relying on the MAGI framework by Yang et al. (2021), physical information is included through a constraint on the residuals of the ODE for the GP solution. The contribution is to improve the scaling of the method by transposing the problem in Fourier space and relying on a Karhunen-Loeve decomposition of the constraint.

**Strengths:**

- the method improve greatly the computational efficiency over the default version
- ablation studies on the effect of truncation are provided

**Weaknesses:**

- spectral decomposition and Fourier features are typical tools for reducing the computational complexity. Some references may be worth adding here: such as Gauthier, B., & Pronzato, L. (2014). Spectral approximation of the IMSE criterion for optimal designs in kernel-based interpolation models. SIAM/ASA Journal on Uncertainty Quantification, 2(1), 805-825., or Mutny, M., & Krause, A. (2018). Efficient high dimensional bayesian optimization with additivity and quadrature fourier features. Advances in Neural Information Processing Systems, 31.
- the state of the art is missing some related works, such as Alvarez, M., Luengo, D., & Lawrence, N. D. (2009, April). Latent force models. In Artificial intelligence and statistics (pp. 9-16). PMLR.

**Questions:**

Can you give the values of W_I?
Can you provide a full pseudo-code for the proposed approach in appendix?
What would be the impact of training the GP hyperparameters including the physical information?

Minor points:
P2: there is redundancy between the first and third paragraphs.
P3: As I understand it, the d-dimensional stochastic process is treated with independent GPs. Then it would be simpler to only put the 1d case here while the extension is direct.
P4L207: missing parenthesis in 2n-1x2n-1
Tables: please put best results in bold
L885: Kramer?
S3.3: what is the value of nu used here? Do you tune it?

---

> ### Author Response · Authors · 2025-11-14
>
> Thanks for your insightful comment for our paper. We will address the concerns and questions you raised.
>
> - **Values of $W_I$**: As defined by [Yang et al, 2021](https://www.pnas.org/doi/10.1073/pnas.2020397118), the $W_I = \max_{t\in I} |\dot{X}(I) - f(X(I),\mathbf\theta,I)|$, where $I = (t_1,t_2,\ldots, t_n)$, is a random variable for quantifying the difference between GP and ODE. By the fact that $W\equiv 0$ if and only if ODEs with parameter $\mathbf \theta$ are satisfied by the Gaussian vector $X(I)$ (i.e., GP $X(t)$ given $I$). Then, the third term of the posterior Eq.(6) is
> $$
> P(W_I = 0 \mid Y(\tau), X(I), \Theta)
> $$
> $$
> = P(\dot{X}(I) - f(x(I),\theta,t_I)=0 \mid X(I), \Theta)
> $$
> $$
> = P(\dot{X}(I) - f(x(I),\theta,t_I)=0 \mid X(I))
> $$
> $$
> = P(\dot{X}(I)=f(x(I),\theta,t_I)\mid X(I)),
> $$
> which is the conditional density of $\dot{X}(I)$ given $X(I)$ evaluated at $f(x(I), \theta, t_I)$. Therefore, $W_I$ is a conceptual variable used to measure the information between the GP and the ODE, rather than a concrete observable quantity.
>
> - **Full pseudocode**: Thanks for your insightful comment about pseudocode for our method. Here is a pseudocode for our method and we will place it in appendix. Due to the characters limit, we have attached a screenshot for your reference [pseudocode](https://ibb.co/JW8VqSnP)
>
> - **Training the GP hyperparameters**: Thanks for your insightful comment about training the GP hyperparameters together with the physical information. This is a very interesting direction for future work. Fortunately, there is a paper by [Besginow et al, 2022](https://arxiv.org/pdf/2208.12515) that explores this idea. The paper presents an approach that automatically incorporates ODE system parameters as GP parameters, learns them during the GP training process, and enables interpretation of the data. We believe this is closely related to GP-based methods, and we plan to include it in our literature review as well.
>
>
> - **Additional References**: We have reviewed the article you provided regarding spectral decomposition and Fourier features, and we will append the following text **to the end of the second paragraph of the literature review**:\
> *Beyond FNO, working in Fourier space has also proven beneficial in other fields. For example, [2] introduces an approximation strategy using deterministic Fourier features for Bayesian optimization, demonstrating both reduced computational cost and exponentially decreasing approximation error. For eigendecomposition, [1] shows that optimizing a truncated IMSE criterion using only a few leading eigenpairs can already yield IMSE-optimal (or near-optimal) designs while significantly reducing computational cost.*
>
>     [1] Gauthier, B., & Pronzato, L. (2014). Spectral approximation of the IMSE criterion for optimal designs in kernel-based interpolation models. SIAM/ASA Journal on Uncertainty Quantification, 2(1), 805-825\
>     [2] Mutny, M., & Krause, A. (2018). Efficient high dimensional bayesian optimization with additivity and quadrature fourier features. Advances in Neural Information Processing Systems, 31
>
>
>     Furthermore, we appreciate your effort in pointing out the missing article. With the inserted discussion we talked about earlier, we will add the following text **to the end of the first paragraph of the literature review**.
>
>     *Furthermore, Gaussian processes (GPs) have been extended to handle decoupled ODE systems. [3] place a GP prior on decoupled ODE systems, which are modeled with constant coefficients and right‐hand‐side forcing functions, in what are known as Latent Force Models (LFMs). This GP prior on the latent forces is then pushed through the differential operator and the and Green’s operator. More recently, [4] introduced a novel class of GPs, LODE-GPs, whose realizations are guaranteed to satisfy exactly a given system of homogeneous linear ODEs with constant coefficients.*
>
>     [3] Alvarez, Mauricio, David Luengo, and Neil D. Lawrence. "Latent force models." Artificial intelligence and statistics. PMLR, 2009.\
>     [4] Besginow, Andreas, and Markus Lange-Hegermann. "Constraining Gaussian processes to systems of linear ordinary differential equations." Advances in Neural Information Processing Systems 35 (2022): 29386-29399.
>
>
>
> - **Minor points**: Thanks for pointing out these minor points in our paper. We will fix the redundant content and typos, and we will use boldface for the best results. Additionally, for the questions you mentioned:
>     - **$\nu$**: The value of the degree of freedom $\nu$ is set to 2.01, as suggested by [Yang et al, 2021](https://www.pnas.org/doi/10.1073/pnas.2020397118)
>     - **GP**: Your understanding that the $d$-dimensional stochastic process is treated with independent GPs is absolutely correct.

---

> ### Comment · Reviewer_ev1R · 2025-11-17
>
> Thank you for your detailed reply. I only have a couple of follow up comments:
> - first bullet point: the idea would be to provide some uncertainty quantification on the difference between GP and ODE, say be returning the probability you detailed (or other quantities).
> - on the spectral decomposition and Fourier features, please note that these are only examples among many others.
>
> Given the scope of the contribution and the comments from other reviewers, I keep my score as is.

---

> > ### Author Response · Authors · 2025-11-29
> >
> > Thanks for your followup comment!
> >
> > - **Uncertainty Quantification:** You are absolutely correct that our method can be extended to provide uncertainty quantification. In future work, we plan to incorporate an MCMC-based procedure to enable full uncertainty quantification.
> >
> > - **examples of spectral decomposition and Fourier features:** Thank you for the suggestion. We have updated the literature review as follows:
> >
> >     *Beyond FNO, the use of Fourier domain representations has been shown to be advantageous in numerous other domains; the examples provided here represent only a subset of a much broader body of work.*
> >
> > We sincerely appreciate your valuable time and thoughtful comments.

---

### Official Review · Reviewer_rg3C · 2025-10-29

**Soundness:** 3
**Presentation:** 3
**Contribution:** 3
**Rating:** 6
**Confidence:** 3

**Summary:**

The paper proposes EFiGP (Eigen–Fourier Physics-Informed Gaussian
Process), an integration-free Bayesian framework for parameter
estimation and trajectory inference in ODE-based dynamical systems.
Building upon the MAGI framework, the authors introduce two
modifications:

*  enforcing the physics-informed constraint in the Fourier domain,
which allows truncation of high-frequency components for denoising
and efficiency; and

*  applying eigendecomposition truncation of the GP covariance to
reduce computational cost.

**Strengths:**

* The paper is well-writen and easy to follow
* The approach is simple yet effective, offering a clear
computational improvement over MAGI.
* Empirical results demonstrate faster inference and stable trajectory recovery.

**Weaknesses:**

* The weakness is that the experimental comparison is somewhat limited in scope. The paper would be strengthened by incorporating additional methods, e.g., non-GP-based methods, to better contextualize the proposed method’s advantages.

**Questions:**

* How scalable is the proposed method for high-dimensional ODE systems?

---

> ### Author Response · Authors · 2025-11-14
>
> Thank you for your insighful comment on our paper. We will address the concerns and questions you raised.
>
> - **More Comparison**: You are absolute correct about more experimental comparisons is helpful for to strengthen our paper. We then compare our method **with the differentiable classical ODE solver**, the Runge–Kutta method, which is a strong baseline suggested by [McGreivy and Hakim, 2024](https://www.nature.com/articles/s42256-024-00897-5). Since $\mathbf{x}(t)$ is always obtained from the ODE solver, the discretization size does not play a key role here. Hence, we compare our EFiGP with the RK method under **different setups by increasing the number of observations** $\mathbf{\tau} = (\tau_1,\tau_2, \ldots, \tau_N)$. For the discretization size, MAGI and EFiGP are both fixed at 1281. We have attached the full comparison results for the absolute parameter errors and trajectory RMSE over 20 runs on the FN system for your reference. From this table, we observe that the performance of inferred trajectories improves as the number of observations increases for all methods. However, our EFiGP consistently outperforms MAGI in both parameter estimation and trajectory recovery. With 41 observations, the RK method achieves an RMSE on $x_1$ that is roughly half that of EFiGP. Yet, with 321 observations, the **inferred trajectories become comparable** to those obtained from the RK method. Due to page limitations, we have included the comparison results on the LV system, as well as the computational cost for both the FN and LV systems, in Appendix S3.1. For the Hes1 system, because of technical issues with the available open-source implementations, RK and other numerical solver based methods were not able to run successfully.
>
>
> **Table: Mean ± SD for FN System (Parameters & Trajectories)**
> | Method    | Param / Traj | **41**      | **81**      | **161**     | **321**     |
> | --------- | ------------ | ----------- | ----------- | ----------- | ----------- |
> | **EFiGP** | a            | 0.031±0.024 | 0.027±0.016 | 0.015±0.008 | 0.017±0.006 |
> |           | b            | 0.233±0.103 | 0.126±0.019 | 0.099±0.025 | 0.086±0.016 |
> |           | c            | 0.050±0.034 | 0.027±0.014 | 0.024±0.008 | 0.019±0.008 |
> |           | $x_1$           | 0.276±0.123 | 0.122±0.042 | 0.104±0.041 | 0.077±0.014 |
> |           | $x_2$          | 0.091±0.038 | 0.041±0.008 | 0.037±0.007 | 0.034±0.010 |
> | **MAGI**  | a            | 0.031±0.018 | 0.036±0.013 | 0.020±0.013 | 0.018±0.011 |
> |           | b            | 0.500±0.085 | 0.458±0.034 | 0.408±0.049 | 0.343±0.050 |
> |           | c            | 0.231±0.080 | 0.209±0.045 | 0.156±0.044 | 0.129±0.050 |
> |           | $x_1$          | 0.427±0.153 | 0.285±0.069 | 0.237±0.042 | 0.184±0.044 |
> |           | $x_2$           | 0.208±0.037 | 0.170±0.014 | 0.154±0.027 | 0.126±0.020 |
> | **RK**    | a            | 0.041±0.023 | 0.019±0.017 | 0.022±0.015 | 0.016±0.011 |
> |           | b            | 0.126±0.010 | 0.109±0.016 | 0.051±0.040 | 0.047±0.031 |
> |           | c            | 0.038±0.022 | 0.021±0.017 | 0.023±0.020 | 0.026±0.014 |
> |           | $x_1$          | 0.127±0.076 | 0.084±0.034 | 0.062±0.035 | 0.057±0.025 |
> |           | $x_2$          | 0.064±0.029 | 0.033±0.008 | 0.022±0.004 | 0.020±0.004 |
>
>
> - **Scalability**: Thanks for your insightful comment on the scalability. Theortically, since all components in our single component formulation remain Gaussian, the full posterior for the log version is obtained by summing over all $D$ components. Consequently, the overall computational cost scales as $O(Dbl)$, where $b$ and $l$ are prefixed. Thus, EFiGP remains **feasible for large scale systems**. To verify this, we also conducted a set of experiments evaluating the computational cost across different state dimensions $D$, including the LV system $(D = 2)$, the Hes1 system $(D = 3)$, and the SEIR system $(D = 4)$, using $j = 81$, $l = 41$, and a discretization of 1281 points. The results are attached below for your reference. These empirical results are consistent with our theoretical $O(D)$ complexity. Although both MAGI and EFiGP exhibit approximately linear growth in runtime, **EFiGP is substantially faster** across all tested systems
>
> **Runtime Comparison**
> | Method / Setting         | **D = 4 (SEIR)** | **D = 3 (Hes1)** | **D = 2 (LV)** |
> | ------------------------ | ---------------- | ---------------- | -------------- |
> | **EFiGP**      | 3.502 ± 0.068    | 2.990 ± 0.090    | 2.100 ± 0.160  |
> | **MAGI**                 | 20.823 ± 1.080   | 15.062 ± 0.429   | 9.714 ± 0.112  |
>
> Thank you for your valuable time, and we hope we have fully addressed all of your concerns.

---

> > ### Comment · Reviewer_rg3C · 2025-11-28
> >
> > Thanks for the reply. I will maintain my rating.

---

### Meta-Review · Area_Chair_AHPT · 2026-01-11

**Summary:**

The reviewers note that the method itself seems reasonable, but that the value of this method lies mostly in how well it performs in practice. As such, excellent experimental evidence is required, and this is currently not the case. Reviewers mention that 1) comparisons are only performed against one other method (MAGI), 2) that predictive performance improvements are small, and 3) that the main issue of computational cost reduction is also modest compared to Fenrir.

To add my own point of view, having worked on dynamical systems in the past: Benchmarking dynamical systems is difficult, and really needs to be done well in order to write a good paper. It is difficult to determine how useful a method is from numerical benchmarks (as are provided in this paper) alone, without additional context. One way to provide context, is to establish a clear problem, for which a particular predictive performance level is needed to solve it.

In engineering journals, there is a whole field of system identification that does this incredibly well. One example of a broader problem, is to find a controller for a dynamical system. The required performance level is then determined by finding to what degree the dynamical system needs to be learned, to be able to find a controller that stabilises the system. This way of benchmarking gives a clear interpretation for how meaningful an improvement in a numerical benchmark is. This is currently lacking in this paper.

In addition, the speed-ups are reported on benchmarks that only take a few seconds. This is hardly a meaningful speedup. If a benchmark only takes a few seconds, computational time is not a prohibitive constraint. A better demonstration would be an experiment that would take hours, if not for this method. In addition, computational time experiments that only take a few seconds are strongly influenced by constant factors that come from implementation details. It is necessary to establish that the method is still beneficial if it were to be scaled to increasingly computationally demanding problems.

**Reviewer Concerns:**

The reviewers engaged well with the authors, but the issues discussed above could not be addressed adequately in the rebuttal period.

**Reviewer Scores:**

There was significant discussion, and it is unlikely that reviewers would have changed their scores significantly to allow acceptance.

---

### Decision · Program_Chairs · 2026-01-26

Reject